# POISONING WITH A PILL: CIRCUMVENTING DETECTION IN FEDERATED LEARNING

## ABSTRACT

Federated learning (FL) protects data privacy by enabling distributed model training without direct access to client data. However, its distributed nature makes it vulnerable to model and data poisoning attacks. While numerous defenses filter malicious clients using statistical metrics, they overlook the role of model redundancy, where not all parameters contribute equally to the model and attack performance. Current attacks manipulate all model parameters uniformly, making them more detectable, while defenses focus on the overall statistics of client updates, leaving gaps for more sophisticated attacks. We propose an attack-agnostic augmentation method to enhance the stealthiness and effectiveness of existing poisoning attacks in FL, exposing flaws in current defenses and highlighting the need for fine-grained FL security. Our three-stage methodology, including *pill construction*, *pill poisoning*, and *pill injection*, injects poison into a compact subnet (*i.e.*, pill) of the global model during the iterative FL training. Experimental results show that FL poisoning attacks enhanced by our method can bypass 8 state-of-the-art (SOTA) defenses, gaining an up to 7x error rate increase, as well as on average a more than 2x error rate increase on both IID and non-IID data, in both cross-silo and cross-device FL systems.

## 1 INTRODUCTION

With the increasing need of machine learning and cloud computing, Federated Learning (FL) (Konečnỳ et al., 2016; McMahan et al., 2017) has become a prominent method for training models using distributed data from numerous clients. Unlike traditional centralized machine learning, FL does not require direct data access, thus reducing communication overhead and preserving data privacy. However, its distributed architecture makes FL vulnerable to attacks if clients are compromised. Many studies (Baruch et al., 2019; Fang et al., 2020; Bhagoji et al., 2019; Shejwalkar & Houmansadr, 2021; Cao & Gong, 2022; Bagdasaryan et al., 2020) have explored these *poisoning attacks*, where malicious clients alter the global model's behavior. These attacks are categorized as: 1) *Model poisoning*, where attackers directly modify local updates to skew global parameters (Fang et al., 2020; Shejwalkar & Houmansadr, 2021); and 2) *Data poisoning*, where malicious samples are injected into local training datasets (Bagdasaryan et al., 2020; Tolpegin et al., 2020; Xie et al., 2020; Sun et al., 2019; Wang et al., 2020; Chen et al., 2017; Liu et al., 2018; Qi et al., 2022). Poisoning attacks pose significant risks to FL (Lyu et al., 2020; Kairouz et al., 2021; Mothukuri et al., 2021; AbdulRahman et al., 2020), undermining its integrity and reliability.

To mitigate these attacks, defenses have been proposed, including *adaptive client filtering*(Blanchard et al., 2017; Cao et al., 2021; Xu et al., 2021; Nguyen et al., 2022; Rieger et al., 2022; Yan et al., 2023b), *statistical parameter aggregation*(Yin et al., 2018; Guerraoui et al., 2018; Fung et al., 2018; Panda et al., 2022), *client-dominant detection*(Guo et al., 2021; 2024; Sun et al., 2021a; Zhang et al., 2023b; Zhu et al., 2023), and *other advanced metrics and pipelines*(Xie et al., 2019; 2021; Cao et al., 2023; 2022; Zhang et al., 2022a). These methods focus on detecting abnormal updates, which are typically obvious in existing attacks, especially in existing model poisoning attacks that treat all neural network parameters uniformly.

We argue that modifying all parameters uniformly is not a cost-effective approach. Studies on model pruning (Frankle & Carbin, 2018; Lin et al., 2018; Han et al., 2015; Mugunthan et al., 2022; Jiang et al., 2022) show that parameters do not contribute equally to a model's performance. Altering

*redundant* parameters wastes resources and reduces attack *stealthiness*. A more effective strategy is to target *critical* parameters (Zhang et al., 2023a), which significantly impact performance, thereby increasing the attack's effectiveness while maintaining stealthiness.

Thus, we propose a novel attack-agnostic augmentation method that enhances model poisoning attacks using a three-stage pipeline: *pill construction*, *pill poisoning*, and *pill injection*. In the first stage, we design a pill blueprint and identify its corresponding subnet instance in the target model. During *pill poisoning*, existing FL attacks are applied in an attack-agnostic manner to poison the selected pill subnet. Finally, in *pill injection*, the poisoned pill is inserted into an estimated benign update, and a two-step adjustment is used to minimize the difference between the poisoned and benign updates. This approach dynamically generates, poisons, and injects a pill into the global model, augmenting existing FL poisoning attacks.

We conduct extensive experiments to evaluate the effectiveness of our augmentation method. We apply it to four baseline poisoning attacks: sign-flipping attack, trim attack (Fang et al., 2020), krum attack (Fang et al., 2020), and min-max attack (Shejwalkar & Houmansadr, 2021). Using both the original and augmented versions, we measure error rates (*i.e.*, the proportion of incorrect predictions) of the global model trained with nine aggregation rules: FedAvg (McMahan et al., 2017), FLTrust (Cao et al., 2021), Multi-Krum (Blanchard et al., 2017), Median (Yin et al., 2018), Trim (Yin et al., 2018), Bulyan (Guerraoui et al., 2018), FLDetector (Zhang et al., 2022a), DnC (Shejwalkar & Houmansadr, 2021), and Flame (Nguyen et al., 2022). These aggregation rules represent most existing defense metrics. We also design an adaptive defense where the defender has full knowledge of our pipeline and implementation. Results show our method substantially improves existing FL poisoning attacks, leading to over a 2x average increase in model prediction error rates under existing defenses, and up to a 7x increase in some cases.

Our contributions are summarized as follows:

- We propose a generic, attack-agnostic augmentation method that enhances poisoning attacks against robust FL by encapsulating model poisoning attacks into well-defined subnets (*i.e.*, pills) with comprehensive metric-based adjustments.

- Extensive experiments on three common datasets against nine aggregation rules demonstrate that our method helps baseline attacks bypass almost all existing defenses, which cannot be successfully attacked by the original baseline attacks.

- We identify limitations of existing poisoning attacks and defenses in FL, highlighting the need and potential for fine-grained FL security.

## 2 BACKGROUND AND RELATED WORK

### 2.1 FEDERATED LEARNING

Federated Learning (FL) (Konečnỳ et al., 2016; McMahan et al., 2017) trains a global model using the information from a swarm of clients without the direct access to each client's data. In a standard FL training process, within an arbitrary communication round $t$, the FL server first distributes its global model $\boldsymbol{g}_t$ to all the clients $K$. After receiving this global model, each client $i$ trains a local model $\boldsymbol{g}_t^{(i)}$ with its local data $D^{(i)}$, and uploads the model update $\Delta \boldsymbol{g}_t^{(i)}$ to the FL server. After receiving the model updates from the clients, the FL server uses aggregation rules to calculate the global model $\boldsymbol{g}_{t+1}$ for the next round. The objective of FL can be formulated as:

$$\min_{\boldsymbol{g}} \sum_{i=0}^{K} \frac{|D^{(i)}|}{|D|} \cdot f(D^{(i)}, \boldsymbol{g}). \tag{1}$$

### 2.2 POISONING ATTACKS IN FL

Based on prior investigations (Shejwalkar et al., 2022; Khan et al., 2023; Jere et al., 2020), existing poisoning attacks in Federated Learning (FL) can be classified into *model poisoning attacks* and *data poisoning attacks*, according to the techniques employed by attackers. In *model poisoning attacks*, attackers may directly compromise the global model by manipulating the updates from local

models (Baruch et al., 2019; Fang et al., 2020; Shejwalkar & Houmansadr, 2021; Cao & Gong, 2022; Bhagoji et al., 2019) by compromised clients. Alternatively, in *data poisoning attacks*, they may poison their local datasets to indirectly influence the global model (Tolpegin et al., 2020; Bagdasaryan et al., 2020; Xie et al., 2020; Sun et al., 2019; Wang et al., 2020; Zhang et al., 2022b). More details of existing FL attacks are presented in Appendix A.1.

Additionally, our method's pill design is inspired by a specialized data poisoning attack known as the *subnet replacement attack* (SRA) (Qi et al., 2022). This approach concentrates backdoor attacks within an arrow-width subnetwork of the original model. It trains this selected subnet using poisoned data and replaces the corresponding parameters of the target model with those from the trained subnetwork. Once the replacement is complete, SRA severs the connections between the poisoned subnetwork and the original model to preserve the efficacy of the attack. The stealthy yet effective design of SRA inspires our method. In particular, we devise a new subnet structure, referred to as the *pill blueprint*, which features heterogeneous widths to better accommodate a variety of existing FL poisoning attacks. Besides, unlike SRA's one-time injection, our method gradually poisons the global model throughout the entire FL training, achieving better effectiveness against a wide range of defenses in the FL setting.

## 2.3 DEFENSES AGAINST POISONING ATTACKS IN FL

Existing defenses can be categorized based on the mitigation strategies that they utilize, including *Adaptive Client Filtering*, *Statistical Parameter Aggregation*, *Client-dominant Detection*, and *Other Advanced Metrics and Pipelines*. More details are presented in Appendix A.2.

To comprhensively evaluate our method, we use *Multi-Krum (MKrum)* (Blanchard et al., 2017), *Trimmed Mean (Trim)* (Yin et al., 2018), *Coordinate-wise Median (Median)* (Yin et al., 2018), *Bulyan* (Guerraoui et al., 2018), *FLTrust* (Cao et al., 2021), *FLDetector (FLD)* (Zhang et al., 2022a), *DnC* (Shejwalkar & Houmansadr, 2021), and *Flame* (Nguyen et al., 2022), a set of representative methods, as our baselines. More details are shown in Appendix A.2.

## 2.4 THREAT MODEL

We follow the typical threat model used in existing studies (Fang et al., 2020; Shejwalkar & Houmansadr, 2021), where the attacker has access to a subset of compromised clients and aims to increase the error rates of the global model on specific classes or across all classes. In this scenario, defenses cannot directly analyze the data on each client as the defender's setting in Blanchard et al. (2017); Yin et al. (2018); Cao et al. (2021); Guo et al. (2021). Instead, they identify malicious clients by analyzing the uploaded client updates. Further details are provided in Appendix B.

## 3 DESIGN OBJECTIVES AND CHALLENGES

After analyzing the drawbacks and various implementations of existing FL poisoning attacks, we define three main objectives for our attack augmentation method: 1) For *stealthiness*, the augmentation method should stay stealthy while achieving comparable performance with original attacks. 2) For *compatibility*, the augmentation should be compatible with most of the existing FL poisoning attacks with few modifications on their implementations. 3) For *generality*, the attack augmentation should be able to bypass general detection methods with different detection metrics.

Corresponding to each objective, three challenges need to be addressed:

- It presents a significant challenge that the attack augmentation method must use significantly fewer parameters while still achieving similar results as the original attacks.

- It is challenging to develop a uniform augmentation method for various FL poisoning attacks since they require different information and are implemented in different training stages.

- It is difficult to devise a general strategy that bypasses all common detection approaches, while guaranteeing the attack effectiveness.

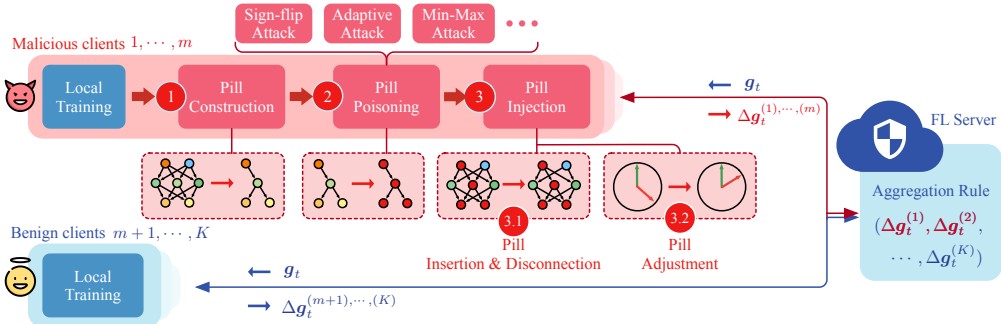

**Figure 1: Overview of our augmentation method. The red parts indicate our augmentation method's contribution, and the cyan parts represent the standard federated learning architecture.**

## 4 DESIGN

**Table 1: Main notations. Symbols in the gray part are used for attacks.**

| Symbol | Meaning |
|---|---|
| $T$ | Total FL communication round |
| $t$ | FL communication round index |
| $K$ | Total client number |
| $g$ | Global model of the FL training |
| $g_t$ | Global model in round $t$ |
| $lr$ | Learning rate |
| $f()$ | Loss function used in the FL training |
| $i$ | Client index |
| $E$ | Local training epoch number |
| $D^{(i)}$ | Local training data on client $i$ |
| $\Delta g_t^{(i)}$ | Local model update of client $i$ in round $t$ |
| $m$ | Total amount of malicious clients |
| $D^m$ | Aggregated data from compromised clients |
| $\Delta \hat{g}_t^m$ | Update of extra-trained model in round $t$ |
| $\Delta \tilde{g}_t$ | Estimated global model update in round $t$ |
| $\Delta g_t^{zero}$ | Disconnection update in round $t$ |
| $M$ | Selected malicious subnetwork |
| $M_{disc}$ | Disconnection mask corresponding to $M$ |
| $C_{iter}$ | Max malicious update adjustment iteration |
| $C_\uparrow$ | Up-scaling factor |
| $C_\downarrow$ | Down-scaling factor |

**Algorithm 1: Our method's workflow. Assume the first $m$ clients as malicious clients.**

```
1  function MalUpdate(i, t, g_t)
      % 1. Pill Construction
2     M, M_disc ← Search(g_t);
      % 2. Pill Poisoning
3     ĝ_{t+1}^m ← g_t;
4     for each epoch e_extra ← 1, ⋯, E_extra do
5        sample B^m from aggregated local data D^m
            on compromised clients;
6        ĝ_{t+1}^m ← ĝ_{t+1}^m − lr · ∇f(B^m, ĝ_{t+1}^m);
7     Δĝ_{t+1}^m ← g_t − ĝ_{t+1}^m;
8     Δg_{t+1}^{(i)} ← Poisoning(i, t, param, Δĝ_{t+1}^m);
9     Δg_{t+1}^{(i)} ← M ⊙ Δg_{t+1}^{(i)};
      % 3. Poison Pill Injection
10    param ← {Δg'_{t+1}^{(1),⋯,(m)}};
11    Δ\tilde{g}_{t+1} ← Estimation(i, t, g_t, param);
12    Δg_{t+1}^{(i)} ← Δg_{t+1}^{(i)} + (1 − M) ⊙ Δ\tilde{g}_{t+1};
13    Δg_{t+1}^{zero} ← 0 − g_t;
14    Δg_{t+1}^{(i)} ← M_disc ⊙ Δg_{t+1}^{zero} + (1 − M_disc) ⊙ Δg_{t+1}^{(i)};
15    param = param ∪ {M_all = M + M_disc};
16    Δg_{t+1}^{(i)} ← SimAdjust(param, Δ\tilde{g}_{t+1}, Δg_{t+1}^{(i)});
17    Δg_{t+1}^{(i)} ← DistAdjust(param, Δ\tilde{g}_{t+1}, Δg_{t+1}^{(i)});
18    return Δg_{t+1}^{(i)}
```

### 4.1 OVERVIEW OF OUR METHOD

Figure 1 presents the three key stages. Table 1 presents notations of main symbols in this paper.

Stage ①: **Pill Construction**. It leverages a dynamic subnetwork search algorithm to achieve *stealthiness* by selecting the poison pill from the global model $g_t$, considering the importance of model's parameters. Since the global model continuously changes across rounds, it is hard to have a fixed pill pattern.

Stage ②: **Pill Poisoning**. In this state, we reapply existing FL poisoning attacks to the selected poison pill, using an extra trained model $\hat{g}_{t+1}^m$ (trained on data from the compromised clients) as the attacker's base model. For *compatibility*, we only modify the input of the existing FL poisoning attacks and utilize their outputs, without any interference to their internal implementations. This black-box utilization lets our method be attack-agnostic and compatible with most of the existing FL poisoning attacks.

Stage ③: **Poison Pill Injection**. It contains poison pill insertion & disconnection, and poison pill adjustment. In this stage, our augmentation method injects the poison pill into the estimated benign update $\Delta \tilde{g}_{t+1}$, and further adjusts the boosting magnitude of both the poison pill parameters and the

remaining parameters. We propose a two-step dynamic adjustment to enhance the *generality* of our method against most defenses.

We are the first that propose a universal attack augmentation pipeline for most FL poisoning attacks, considering *stealthiness*, *compatibility*, and *generality*. The detailed workflow of our method is shown in Algorithm 1.

## 4.2 Pill Construction

This stage aims to construct a pill structure for augmenting the *stealthiness* while retaining the attack effectiveness before being augmented. The pill is carefully crafted to involve a minimal subset of parameters from specific positions of the target model. We first define a pill's blueprint as the pill's graphic structure, independent of target model parameters. Then, we propose a dynamic pill search algorithm to identify and map concrete parameters from the target model to the blueprint.

**Designing Pill Blueprint.** The blueprint design is inspired by SRA (Qi et al., 2022), which shows that poisoning a narrow subnetwork (one neuron/channel in each layer) is adequate to effectively inject backdoors into machine learning models (not in the FL setting). However, their technique cannot be used for our purposes as their subnet architecture is very specific. It does not support attacking various targets; it is a fixed and pre-selected subnet without considering the dynamics of model training in FL; and its poisoned subnet is not stealthy, having substantially larger weight values compared to others due to the need to disseminate the poison effect through such a small pre-selected network. Therefore, we propose a novel blueprint method, in which the subnet structure is general, and its instantiations (i.e., the concrete subnets) vary across steps in the FL training procedure, including important neurons by a dynamic search algorithm. This allows small weight changes because poisoning important neurons enables easy dissemination, maximizing attack stealthiness. In particular, the pill blueprint is designed to accommodate various target classes of different FL poisoning attacks. It achieves this by manipulating the outputs relevant to multiple classes simultaneously, via disrupting all the output neurons together. Hence, our pill blueprint design follows the rules below:

1. The pill blueprint only contains one neuron in each linear layer or one channel in each convolutional layer, except for the last two layers. Suppose $\mathcal{N}_i^p$ represents the neuron/channel number in Layer $i$ in our pill blueprint, then $\mathcal{N}_i^p = 1$ when $i < L - 1$, where $L$ is the total layer counts in our pill blueprint.

2. In the last two layers of our pill blueprint, $\mathcal{N}_{L-1}^p = \mathcal{N}_L^p = $ *number of classes*.

**Dynamic Pill Search.** According to existing studies on neural network pruning (Frankle & Carbin, 2018; Lin et al., 2018; Han et al., 2015; Mugunthan et al., 2022; Jiang et al., 2022), parameters with a larger magnitude typically dominate the model's performance. The optimal solution is hence to examine the model parameters to search for a globally optimal pill that encompasses the most important parameters.

However, such a globally optimal pill could be identified via a pruning-based method (Wu et al., 2020; Sun et al., 2021b), and hence our attack could be easily detected. Besides, searching for a globally optimal pill is inefficient when the model has a large number of parameters. Thus, we search for an approximate pill instead, with an attacker-defined start point, and only evaluate a small subset of the entire model's parameters. We name the search algorithm as "approximate max pill search". The key idea is to perform a targeted neuron search at each layer by focusing only on the neurons connected to the selected neurons from the previous layer, following a high-sum-of-weights-first principle that prioritizes neurons based on the cumulative sum of their connection weights to the previously selected neurons. The entire search contains four steps:

**Step 1 Random Start Point Selection:** Randomly select neurons from the first layer of the target model, denoted as $\mathcal{V}_1$, based on the pill's blueprint and neuron count $\mathcal{N}_1^p$ in its first layer. These neurons are fixed as start points throughout the search.

**Step 2 Layer-wise Search:** For each subsequent layer $l_i$, we compute the sum of weights connecting neurons in $\mathcal{V}_{i-1}$ to neurons in $l_i$ and rank the neurons in $l_i$ based on the descending order of the sum of weights. Top $\mathcal{N}_i^p$ neurons are chosen for $\mathcal{V}_i$.

**Step 3 Output Neuron Pairing:** Pair the selected neurons $\mathcal{V}_{L-1}$ in the final hidden layer with the neurons in the output layer $l_L$, ensuring a one-to-one correspondence.

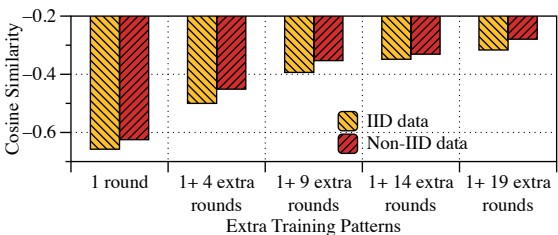 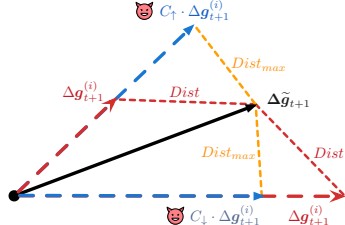

Figure 2: Cosine similarities between FLTrust server's model update and malicious model update when malicious clients use different extra training rounds.

Figure 3: Intuition behind distance-based adjustment in our augmentation method.

**Step 4 Pill Mask Construction:** Two masks are constructed. $M$ marks the pill parameters in the target model, and $M_{disc}$ records the disconnection locations between the pill and the remained neurons in the target model.

The searched pill ensures both effectiveness and stealthiness when used in attacks. Detailed information for each step is provided in Appendix D, along with a concrete example and an overhead analysis of the pill search algorithm.

### 4.3 PILL POISONING

In the **pill poisoning** stage, we aim to condense the poison into the pill using existing attacks. To achieve compatibility, our method simply reuses existing FL poisoning attacks, without any intrusive modification to their original implementations. We only modify the input of existing FL poisoning attacks by replacing the base model update with the update from a model that has undergone extra training rounds, denoted as $\Delta \hat{g}_{t+1}^{m}$. Additionally, we restrict changes to parameters within the pill. The output is a poisoned pill that will be used in the next **pill injection** stage.

The motivation to use an extra-trained model update as the reference model update is shown in Figure 2. As shown in the figure, with the increasing number of extra training rounds on the malicious clients, the generated malicious model update becomes less opposite to the FLTrust (Cao et al., 2021) server's model update. Thus, we adopt the extra training in our method and limit the extra training epoch number $E_{extra}$ to less than the number of malicious clients $m$ times the benign local training epoch number $E$, denoted as $E_{extra} \leq m \cdot E$. With this extra training epoch number limit, we do not violate the threat model since the attacker can utilize the data and computational resources of all the compromised clients.

### 4.4 PILL INJECTION

In the **pill injection** stage, we aim to inject the pill into the model and use a two-step adjustment method to further camouflage the pill. Thus, the entire injection stage could be divided into two parts - pill injection and camouflaging. After this stage, the poison pill is seamlessly integrated with the benign model update and uploaded to the FL server.

**Pill Insertion & Disconnection.** In this part, our goal is to insert the pill into the model, and minimize the impact of the benign model updates on our pill. We use an estimated global model update as the benign model update, which is estimated as the coordinate-wise mean values of all the normal model updates from the compromised clients. The estimation process in the `Estimation()` in Algorithm 1 `Line 11` is hence presented as Equations (2),

$$\Delta \widetilde{g}_{t+1} \leftarrow mean\{\Delta g_{t+1}'^{(1),\cdots,(m)}\}, \tag{2}$$

where $\Delta g_{t+1}'^{(i)}$ is the normal updates from the compromised clients. By aggregating information from multiple malicious clients, the estimated benign global model update is more similar to the genuine one, providing more budget for our poison pill.

After obtaining the estimated global model update $\Delta \widetilde{g}_{t+1}$, we directly replace the parameters corresponding to the pill parameters (which have been poisoned in the previous stage) via the pill's

mask $M$. Then, we replace the parameters that connect the pill and the other estimated global model updates with *the disconnection update* $\Delta g_{t+1}^{zero}$, using the disconnection mask $M_{disc}$. The disconnection update $\Delta g_{t+1}^{zero}$ is calculated as $0 - g_t$, and is bounded by the maximum and minimum values of the reference model update $\Delta \hat{g}_{t+1}^m$. The disconnection update can gradually change the parameters of the connections between the pill and the rest of the model to $0$, and finally isolates the poison pill from the global model, guaranteeing the attacking effects of the poison pill.

**Pill Adjustment.** After the injection, we use a two-step adjustment to further adjust the pill, improving the generality against multiple detection metrics simultaneously. In this stage, we consider two prevailing detection metrics – distance and cosine similarity. To increase the cosine similarity between the poisoned model update and the benign model update in our method, we balance the magnitudes of both the poison pill's parameters and the other benign parameters. Similarly, to minimize the distance discrepancy between the poisoned and benign model updates, we adjust the magnitude of the entire poisoned model update, as shown in Figure 3. Thus, we first use the **similarity-based adjustment**, then use the **distance-based adjustment**, balancing the *effectiveness* and the *stealthiness* of the poisoned model update. This two-step adjustment is particularly effective when combined with our method, which selectively poisons only a tiny subset of the model's parameters. By altering just a few parameters, our method preserves a substantial number of benign parameters, which are crucial for making effective adjustments. As a result, the poisoned model update can bypass a wide range of defenses since they are typically designed based on the combination or variants of distance and cosine similarity metrics, and they usually do not anticipate such a focused and minimal interference in the model parameters. The details of the two seamless adjustments are shown in Appendix F.

## 5 EVALUATION

This section assesses how our method enhances the effectiveness of existing FL poisoning attacks from multiple perspectives. We begin by evaluating the *Augmentation Effectiveness* of our method against four FL poisoning attacks, using nine prevailing defenses across three datasets, detailed in Section 5.2. Subsequently, we visualize the *Stealthiness* of our method under two prevailing detection metrics, as discussed in Section 5.3. Lastly, the *Generality Analysis* of our method is presented, which includes tests on various proportions of malicious clients, tests on both cross-silo and cross-device settings, and evaluates the impact of different pill search rules, outlined in Section 5.4. Our method significantly enhances the capabilities of existing FL poisoning attacks, successfully bypassing all 9 baseline defenses in over 90% of cases, and increasing the error rates by up to seven times compared to the original attacks' error rates. Moreover, it demonstrates robustness across varying data distributions, model architectures, proportions of malicious clients, and pill search rules.

### 5.1 EVALUATION SETTINGS

In our experiments, we typically set the malicious proportion to $20\%$. We implement 9 baseline aggregation rules, including FedAvg, FLTrust, Multi-Krum, Bulyan, Median, Trim, FLDetector, DnC, and Flame. We use our method to augment $4$ existing model poisoning attacks, including sign-flipping attack, Trim attack, Krum attack, and Min-Max attack. These attacks are chosen for their representativeness in illustrating the effectiveness of our method. We configure a 50-client FL system for both the MNIST and Fashion-MNIST datasets. For the CIFAR-10 dataset, a 30-client FL system is used. Our framework accommodates both cross-silo and cross-device settings. The entire framework is based on PyTorch (Paszke et al., 2019). We present more experimental configurations in Appendix G.

### 5.2 AUGMENTATION EFFECTIVENESS

In this section, we present a comprehensive analysis of our method's augmentation effectiveness on Fashion-MNIST dataset within a 50-client cross-silo FL system, in which 20% clients are malicious. We evaluate our method on both IID data and non-IID data. Our method successfully augments all the baseline attacks with more than $0.25$ average error rate increase, showing our method's *effectiveness* and high *compatibility*.

**Table 2: Error rates under cross-silo setting using "approximate max pill search" (20% malicious clients) on Fashion-MNIST dataset.**

| Data Distribution | IID | | | | | | | Non-IID | | | | | | |
|---|---|---|---|---|---|---|---|---|---|---|---|---|---|---|
| Attack | FedAvg | FLTrust | MKrum | Bulyan | Median | Trim | FLD | FedAvg | FLTrust | MKrum | Bulyan | Median | Trim | FLD |
| No Attack | 0.109 ±0.003 | 0.107 ±0.003 | 0.105 ±0.002 | 0.105 ±0.001 | 0.123 ±0.004 | 0.106 ±0.002 | 0.115 ±0.002 | 0.113 ±0.002 | 0.115 ±0.003 | 0.115 ±0.004 | 0.112 ±0.003 | 0.142 ±0.003 | 0.115 ±0.003 | 0.122 ±0.003 |
| Sign-flipping Attack | **0.943** ±**0.023** | 0.114 ±0.003 | 0.108 ±0.002 | 0.126 ±0.001 | 0.136 ±0.002 | 0.116 ±0.001 | 0.118 ±0.003 | **0.917** ±**0.020** | **0.126** ±**0.004** | 0.117 ±0.002 | 0.132 ±0.003 | 0.152 ±0.006 | 0.124 ±0.003 | 0.127 ±0.003 |
| + Poison Pill | 0.667 ±0.089 | 0.115 ±**0.004** | 0.764 ±**0.049** | 0.379 ±**0.104** | 0.523 ±**0.091** | 0.314 ±**0.018** | 0.646 ±**0.061** | 0.543 ±0.150 | 0.122 ±0.006 | 0.754 ±**0.129** | 0.430 ±**0.057** | 0.522 ±**0.038** | 0.311 ±**0.038** | 0.688 ±**0.067** |
| Trim Attack | 0.243 ±0.010 | 0.109 ±0.003 | 0.139 ±0.002 | 0.146 ±0.006 | 0.174 ±0.006 | 0.179 ±0.003 | **0.116** ±**0.001** | 0.332 ±0.022 | 0.120 ±0.005 | 0.201 ±0.018 | 0.163 ±0.004 | 0.231 ±0.008 | **0.238** ±**0.009** | 0.124 ±0.003 |
| + Poison Pill | **0.618** ±**0.071** | **0.576** ±**0.057** | **0.638** ±**0.041** | **0.284** ±**0.040** | **0.453** ±**0.091** | **0.219** ±**0.010** | 0.115 ±0.003 | **0.668** ±**0.033** | **0.517** ±**0.038** | **0.687** ±**0.036** | **0.292** ±**0.047** | **0.473** ±**0.047** | 0.223 ±**0.016** | **0.222** ±**0.128** |
| Krum Attack | 0.116 ±0.002 | 0.109 ±0.003 | 0.189 ±0.022 | 0.201 ±0.009 | 0.172 ±0.008 | 0.137 ±0.003 | **0.786** ±**0.087** | 0.128 ±0.004 | 0.116 ±0.003 | 0.235 ±0.059 | 0.276 ±0.003 | 0.217 ±0.005 | 0.160 ±0.003 | **0.947** ±**0.030** |
| + Poison Pill | **0.735** ±**0.032** | **0.155** ±**0.032** | **0.715** ±**0.132** | **0.422** ±**0.046** | **0.578** ±**0.057** | **0.310** ±**0.009** | 0.637 ±0.074 | **0.716** ±**0.104** | **0.151** ±**0.004** | **0.737** ±**0.078** | **0.468** ±**0.017** | **0.730** ±**0.168** | **0.334** ±**0.031** | 0.690 ±0.079 |
| Min-Max Attack | 0.183 ±0.008 | 0.110 ±0.002 | 0.431 ±0.029 | **0.330** ±**0.015** | 0.183 ±0.009 | 0.218 ±0.009 | **0.825** ±**0.052** | 0.269 ±0.026 | 0.125 ±0.015 | 0.619 ±**0.050** | 0.434 ±**0.080** | 0.255 ±0.012 | 0.278 ±0.007 | **0.831** ±**0.049** |
| + Poison Pill | **0.702** ±**0.114** | **0.303** ±**0.201** | **0.668** ±**0.116** | 0.327 ±0.074 | **0.514** ±**0.053** | **0.314** ±**0.047** | 0.778 ±0.063 | **0.629** ±**0.114** | **0.320** ±**0.115** | 0.612 ±0.040 | 0.406 ±0.065 | **0.547** ±**0.072** | **0.376** ±**0.119** | 0.822 ±0.036 |

**Table 3: Error rates under cross-silo setting using "approximate max pill search" (10% malicious clients) on Fashion-MNIST dataset.**

| Data Distribution | IID | | | | | | | Non-IID | | | | | | |
|---|---|---|---|---|---|---|---|---|---|---|---|---|---|---|
| Attack | FedAvg | FLTrust | MKrum | Bulyan | Median | Trim | FLD | FedAvg | FLTrust | MKrum | Bulyan | Median | Trim | FLD |
| No Attack | 0.106 ± 0.003 | 0.104 ±0.003 | 0.103 ±0.003 | 0.108 ±0.004 | 0.127 ±0.001 | 0.107 ±0.002 | 0.116 ±0.002 | 0.111 ±0.002 | 0.119 ±0.003 | 0.113 ±0.001 | 0.113 ±0.002 | 0.140 ±0.005 | 0.114 ±0.002 | 0.123 ±0.004 |
| Sign-flipping Attack | **0.964** ±**0.017** | 0.109 ±0.003 | 0.108 ±0.003 | 0.110 ±0.003 | 0.130 ±0.005 | 0.108 ±0.001 | 0.117 ±0.005 | **0.909** ±**0.045** | 0.119 ±0.002 | 0.114 ±0.003 | 0.119 ±0.002 | 0.144 ±0.004 | 0.120 ±0.004 | 0.125 ±0.002 |
| + Poison Pill | 0.320 ±0.080 | **0.116** ±**0.007** | **0.162** ±**0.027** | **0.151** ±**0.010** | **0.323** ±**0.029** | **0.148** ±**0.007** | **0.699** ±**0.082** | 0.269 ±0.174 | **0.120** ±**0.003** | **0.239** ±**0.101** | **0.164** ±**0.013** | **0.364** ±**0.031** | **0.168** ±**0.012** | **0.242** ±**0.198** |
| Trim Attack | 0.112 ±0.002 | 0.111 ±0.005 | 0.111 ±0.004 | 0.115 ±0.003 | 0.132 ±0.004 | 0.114 ±0.003 | 0.116 ±0.001 | 0.125 ±0.003 | 0.115 ±0.006 | 0.121 ±0.001 | 0.125 ±0.004 | 0.153 ±0.005 | 0.122 ±0.002 | 0.122 ±0.002 |
| + Poison Pill | **0.508** ±**0.128** | **0.139** ±**0.012** | **0.334** ±**0.120** | **0.126** ±**0.006** | **0.284** ±**0.040** | **0.127** ±**0.004** | **0.120** ±**0.003** | **0.528** ±**0.051** | **0.148** ±**0.018** | **0.455** ±**0.151** | **0.143** ±**0.003** | **0.287** ±**0.023** | **0.146** ±**0.004** | **0.136** ±**0.012** |
| Krum Attack | 0.107 ±0.004 | 0.108 ±0.005 | 0.114 ±0.001 | 0.123 ±0.002 | 0.141 ±0.004 | 0.112 ±0.004 | **0.668** ±**0.134** | 0.116 ±0.003 | 0.117 ±0.003 | 0.124 ±0.003 | 0.138 ±0.001 | 0.173 ±0.005 | 0.122 ±0.003 | 0.410 ±0.352 |
| + Poison Pill | **0.183** ±**0.039** | **0.118** ±**0.003** | **0.283** ±**0.210** | **0.161** ±**0.015** | **0.362** ±**0.027** | **0.146** ±**0.008** | 0.631 ±0.089 | **0.428** ±**0.190** | **0.127** ±**0.005** | **0.280** ±**0.064** | **0.187** ±**0.012** | **0.415** ±**0.057** | **0.182** ±**0.009** | **0.704** ±**0.091** |
| Min-Max Attack | 0.117 ±0.004 | 0.108 ±0.004 | 0.118 ±0.002 | 0.135 ±0.005 | 0.142 ±0.009 | 0.128 ±0.008 | 0.111 ±0.004 | 0.124 ±0.003 | 0.119 ±0.012 | 0.142 ±0.003 | **0.166** ±**0.007** | 0.162 ±0.004 | 0.145 ±0.004 | 0.136 ±0.002 |
| + Poison Pill | **0.439** ±**0.140** | **0.129** ±**0.009** | **0.361** ±**0.245** | **0.136** ±**0.015** | **0.343** ±**0.032** | **0.150** ±**0.006** | **0.715** ±**0.096** | **0.521** ±**0.073** | **0.136** ±**0.011** | **0.339** ±**0.202** | 0.153 ±0.009 | **0.368** ±**0.048** | **0.184** ±**0.017** | 0.335 ±0.185 |

**Results on IID Data.** The error rates of four baseline FL poisoning attacks, with and without our method, are shown in the left half of Table 2. Our method enhances the error rates of the existing poisoning attacks in 23 out of 28 scenarios, against FedAvg and five defenses. The maximum increase in error rate is $0.658$, and the average increase reaches $0.274$. This substantial elevation from the attack-free baseline error rate of $0.109$ underscores our method's capability to significantly compromise existing defenses' integrity.

**Results on Non-IID Data.** Evaluations on non-IID data further validate the effectiveness of our method, demonstrating its superiority in 23 out of 28 cases. The highest error rate increase reaches $0.637$, with an average increase of $0.281$. Although there is a slight reduction in maximal error rate increase in the non-IID data setting, these results still demonstrate our method's ability to effectively enhance attacks in more complex and heterogeneous data environments.

All attacks augmented by our method can bypass all baseline defenses, including FLTrust and FLDetector, with the exception of the sign-flipping attack. Notably, the Min-Max attack demonstrates

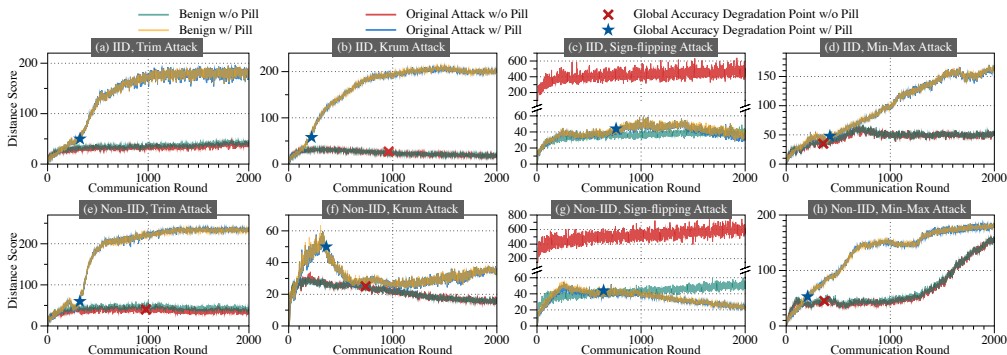

**Figure 4: Comparison of Multi-Krum distance score between benign updates and malicious updates when using original poisoning attacks with and without our method.**

**Table 4: Error rates under cross-silo setting using "approximate max pill search" (20% malicious clients) on CIFAR-10 dataset.**

| Data Distribution | IID | | | | | | | | | Non-IID | | | | | | | | |
|---|---|---|---|---|---|---|---|---|---|---|---|---|---|---|---|---|---|---|
| Attack | FedAvg | FLTrust | MKrum | Bulyan | Median | Trim | DnC | FLD | Flame | FedAvg | FLTrust | MKrum | Bulyan | Median | Trim | DnC | FLD | Flame |
| No Attack | 0.488 | 0.480 | 0.507 | 0.469 | 0.551 | 0.456 | 0.445 | 0.494 | 0.491 | 0.486 | 0.474 | 0.499 | 0.498 | 0.581 | 0.502 | 0.463 | 0.506 | 0.532 |
| Sign-flip Attack | **0.898** | 0.479 | 0.580 | 0.539 | 0.621 | 0.461 | 0.468 | 0.497 | 0.509 | **0.905** | 0.514 | 0.511 | 0.622 | 0.658 | 0.573 | 0.502 | 0.603 | 0.533 |
| + Poison Pill | 0.739 | **0.880** | **0.929** | **0.694** | **0.707** | **0.699** | **0.536** | **0.899** | **0.706** | 0.879 | **0.861** | **0.898** | **0.677** | **0.766** | **0.688** | **0.566** | **0.900** | **0.675** |
| Trim Attack | 0.482 | 0.509 | 0.489 | 0.536 | 0.623 | 0.514 | 0.456 | 0.459 | 0.501 | 0.571 | 0.493 | 0.608 | 0.595 | 0.653 | 0.549 | 0.481 | 0.482 | 0.506 |
| + Poison Pill | **0.853** | **0.877** | **0.883** | **0.654** | **0.674** | **0.662** | **0.513** | **0.899** | **0.542** | **0.890** | **0.862** | **0.906** | **0.772** | **0.688** | **0.639** | **0.518** | **0.893** | **0.621** |
| Krum Attack | 0.473 | 0.541 | 0.471 | 0.568 | 0.540 | 0.510 | 0.455 | 0.802 | 0.500 | 0.485 | 0.506 | 0.497 | 0.522 | 0.647 | 0.519 | 0.481 | **0.899** | 0.501 |
| + Poison Pill | **0.701** | **0.896** | **0.900** | **0.765** | **0.756** | **0.643** | **0.529** | **0.890** | **0.872** | **0.724** | **0.849** | **0.900** | **0.675** | **0.748** | **0.647** | **0.580** | 0.885 | **0.873** |
| Min-Max Attack | 0.450 | 0.504 | 0.469 | 0.507 | 0.579 | 0.465 | 0.514 | 0.525 | 0.525 | 0.478 | 0.502 | 0.493 | 0.568 | 0.636 | 0.603 | 0.478 | 0.482 | 0.488 |
| + Poison Pill | **0.752** | **0.712** | **0.902** | **0.775** | **0.802** | **0.640** | **0.545** | **0.902** | **0.811** | **0.661** | **0.646** | **0.886** | **0.674** | **0.783** | **0.661** | **0.527** | **0.907** | **0.799** |

superior effectiveness in non-IID data settings, achieving significant improvements compared to its performance on IID data. Other attacks also exhibit similar error rate improvements relative to their results on IID data, indicating that our method maintains its robustness and effectiveness in more complex data environments. More detailed analyses are presented in Appendix H.

### 5.3 STEALTHINESS ANALYSIS

To further analyze the performance of our method, we analyze its stealthiness during the training process of the FL system, focusing on how our method influences the distance scores and cosine similarity scores of existing FL poisoning attacks. The results indicate that our method can make malicious clients appear as benign or even more "benign" than genuine benign clients. This significant increase in *stealthiness* is a result of the pill design with distance-based adjustment and similarity-based adjustment techniques in our method. Figure 4 compares the average distance scores of benign and malicious clients (with and without our method) across four baseline model poisoning attacks. The distance scores when using our method closely match or are even identical to those of benign clients throughout the entire training process. Detailed analyses are included in Appendix I, where we also show the results on the similarity scores.

### 5.4 GENERALITY ANALYSIS

In this section, we further discuss the *generality* of our method from four perspectives: malicious client proportion, client participation frequency, datasets & model architectures, and pill search algorithm. The results indicate that our method maintains its augmentation effectiveness consistently, even as these conditions change, demonstrating its reliability and wide applicability in augmenting FL poisoning attacks.

**Impact of The Malicious Client's Proportion.** We first assess the effectiveness of our method in both IID and non-IID cross-silo FL systems with only 10% of clients compromised, as shown in Table 3. This setup reveals that all baseline model poisoning attacks yield lower error rates on the global model compared with scenarios with 20% compromised clients. While the increase in error rates is less than those in the 20% compromised client scenario, our method still effectively raises the global model's error rates in 25/26 out of 28 cases (IID/non-IID setting). The maximum increase in error rates reaches 0.403, with an average increase of 0.144. This average is notably higher (>2x higher) than the error rates observed in attack-free FL conditions. Specifically, our method helps sign-flipping/Trim/Krum/Min-Max attacks achieve an average error rate increase of 0.133/0.094/0.136/0.209. More detailed results are presented in Appendix L.

**Impact of The Client Participation Frequency.** We then extend the evaluation of our method to a cross-device FL system, where only 40% of clients are selected for participation in each communication round. This setup results in less frequent participation from each client and a fluctuating proportion of malicious clients across different rounds. The maximum error rate increase with our method is 0.639, with an average increase across different attacks and defenses of 0.279. These results are consistent with those from the cross-silo FL system, underscoring our method's effectiveness and *generality* across different FL configurations. This evaluation demonstrates our method's robust performance and adaptability, not only in a controlled cross-silo environment but also under the more various conditions in cross-device FL systems. More details are presented in Appendix K.

**Impact of The Datasets and Model Architectures.** Following the evaluation with the Fashion-MNIST dataset, we test our method on the MNIST and CIFAR-10 dataset, employing the four-layer CNN model and the AlexNet model to further verify our method's *generality* across different datasets. The collective results show that our method performs even better with larger datasets or more complex machine learning models. This trend confirms the *generality* of our method by revealing its capability to maintain consistent performance enhancements regardless of the dataset or model complexity involved. Specifically, our method helps all four baseline attacks bypass all nine baselines on CIFAR-10 dataset, achieving 0.288 error rate increase on average, presented in Table 4. More detailed results on MNIST dataset are shown in Appendix J.

We also explore the impact of the pill search algorithm in our method in Appendix M. The results show that the "approximate max pill search" algorithm outperforms the "approximate min pill search" in 41 out of 56 cases (approximately 73%), underscoring its effectiveness in leveraging the most influential parameters to enhance attack impacts. Additional results on ablation studies and generalizability are presented in Appendix P to Appendix U.

## 6    DISCUSSION

To further evaluate the robustness of our method when defenses are aware of the attack strategies (white-box scenario), we design an adaptive defense and present the experimental details in Appendix N. Despite the adaptive defense's attempt to incorporate both cosine similarity and distance metrics, it remains insufficient to thwart the enhanced capabilities of our method. We also presented a detailed discussion of the limitations of our work and future directions in Appendix O.

## 7    CONCLUSION

In this paper, we propose a novel attack-agnostic augmentation method to enhance existing poisoning attacks in FL by concentrating attacks into a pill (a tiny subnet). Our approach is constructed with three stages, including *pill construction*, *pill poisoning*, and *pill injection*. Accordingly, we first use a dynamic pill search algorithm to determine the concrete pill based on the pill blueprint. Then we poison the pill using existing FL poisoning attacks, and carefully inject the poison pill into the target model with two pill-related masks and a two-step adjustment. Our method enables existing poisoning attacks to achieve more than 2x error rates on average compared with their original implementations. The effectiveness of our method in exploiting and exacerbating the inherent weaknesses of current FL defenses highlights the critical need for more refined detection measures in FL.

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

## A    ADDITIONAL DETAILS OF EXISTING ATTACKS AND DEFENSES

### A.1    ADDITIONAL DETAILS OF EXISTING ATTACKS IN FL

Although model poisoning attacks are effective, existing attacks have limited *stealthiness* and can be detected by many existing defenses. Our goal is hence to demonstrate that such attacks can be augmented in a uniform way. Model poisoning attacks directly manipulate the parameters uploaded by clients, with a minimal interference to the local training process. Among these attacks, the simplest form is the *sign-flipping attack*, which directly flips the model update and scales it with a constant factor. *A-Little-is-Enough* (Baruch et al., 2019) generates malicious updates within a calculated perturbation range to deceive the global model. Adaptive attacks (Fang et al., 2020), such as the Trim attack and the Krum attack, dynamically scale malicious updates based on parameter values and distances. The Min-Max and Min-Sum Attacks (Shejwalkar & Houmansadr, 2021) provide a dynamic scaling for malicious updates based on different distance-based criteria. MPAF (Cao & Gong, 2022) aims to drive the global model towards a predefined target model with poor performance on given FL tasks. For the sake of generality, we employ the sign-flipping attack, two types of adaptive attacks (Fang et al., 2020) (the Trim attack and the Krum attack), and the Min-Max attack (Shejwalkar & Houmansadr, 2021) as the baseline attacks in this paper.

### A.2    ADDITIONAL DETAILS OF EXISTING DEFENSES IN FL

*Adaptive Client Filtering.* These techniques such as Krum and Multi-Krum (Blanchard et al., 2017) filter out malicious clients through single or multiple rounds of client selection based on distance scores. FLTrust (Cao et al., 2021) computes trust scores using the cosine similarity between each client update and the server model's update for weighted averaging. SignGuard (Xu et al., 2021) employs sign-based clustering combined with norm-based thresholding to identify and filter malicious clients. Flame (Nguyen et al., 2022) and Deepsight (Rieger et al., 2022) propose adaptive clustering and clipping to safeguard against backdoor attacks. SkyMask (Yan et al., 2023b) clusters trainable feature masks of clients to assess each client's risk level.

*Statistical Parameter Aggregation.* Approaches like Median and Trim (Yin et al., 2018) use coordinate-wise median or trimmed mean values to aggregate model updates. Bulyan (Guerraoui et al., 2018) enhances robustness by integrating Krum with Trim techniques. Fool's Gold (Fung et al., 2018) applies an adaptive learning rate based on inter-client contribution similarity to mitigate the effects of malicious updates. SparseFed (Panda et al., 2022) aggregates sparsified updates, reducing the risk of model poisoning attacks.

*Client-dominant Detection.* Siren (Guo et al., 2021) and Siren$^+$ (Guo et al., 2024) set proactive accuracy-based alarms at the client level with the corresponding server-side decisions to counter various model poisoning attacks. FL-WBC (Sun et al., 2021a) introduces client-side noise to diminish the efficacy of attacks and shorten their duration. FLIP (Zhang et al., 2023b) achieves higher robustness through client-side reverse-engineering defenses against extensive poisoning strategies. LeadFL (Zhu et al., 2023) uses a client-side Hessian matrix optimization to reduce the impact of adversarial patterns on backdoor and targeted attacks.

*Other Advanced Metrics and Pipelines.* Various studies employ other sophisticated metric pipelines designed for detection to ensure robust defense against poisoning attacks. These include techniques proposed in studies such as Zeno (Xie et al., 2019), CRFL (Xie et al., 2021), FedRecover (Cao et al., 2023), FLCert (Cao et al., 2022), FLDetector (Zhang et al., 2022a), and MESAS (Krauß & Dmitrienko, 2023).

Here are more details of the baseline defenses used in our paper:

**Krum and Multi-Krum (MKrum) (Blanchard et al., 2017).** Krum uses a distance score as the metric. In each round, the Krum server sums the distances between each client update $g_t^{(i)}$ and its $K - m - 2$ neighbors, and uses these sums as the scores for all the clients. The Krum server then selects the client's model update with the lowest score. Multi-Krum is a variant of Krum that uses iterative Krum to pick multiple candidates for aggregation.

**Coordinate-wise Median (Median)** (Yin et al., 2018). Coordinate-wise Median (Median) uses the per-parameter median values of the model updates from the clients as the aggregated global model update, which is then used to generate the next-round global model.

**Trimmed Mean (Trim)** (Yin et al., 2018). Trimmed Mean (Trim) calculates per-parameter trimmed mean values of the client model updates and packs them as the global model update.

**Bulyan** (Guerraoui et al., 2018). Bulyan is a combination of Krum and Trim. It first uses the Krum-based method to select multiple candidates, and uses the per-parameter trimmed mean values of the candidate model updates as the final global model update.

**FLTrust** (Cao et al., 2021). FLTrust trains a server model with a small root dataset. In each round, it computes the clipped cosine similarities between the server model update and client updates as trust scores, and then uses the trust scores as weights to aggregate all the normalized client model updates.

**FLDetector (FLD)** (Zhang et al., 2022a). FLDetector filters out malicious clients by checking the multi-round consistency of all client updates. Malicious updates typically have lower consistency compared to benign ones.

**Flame** (Nguyen et al., 2022). Flame utilizes HDBSCAN-based (Campello et al., 2013) dynamic clustering to filter out malicious clients, and aggregates median-clipped benign updates with adaptive noise as the global model update.

Table 5: Architectures of the original CNN model and the corresponding pill blueprint.

| Layer Type | Original CNN Model | Our Pill Blueprint |
|---|---|---|
| Input | $28 \times 28 \times 1$ | $28 \times 28 \times 1$ |
| Conv2d | $3 \times 3 \times 30$ | $3 \times 3 \times 1$ |
| ReLU | -[1] | - |
| MaxPool2d | $2 \times 2$ | $2 \times 2$ |
| Conv2d | $3 \times 3 \times 50$ | $3 \times 3 \times 1$ |
| ReLU | - | - |
| MaxPool2d | $2 \times 2$ | $2 \times 2$ |
| Linear | $1250 \times 100$ | $25 \times 10$ |
| ReLU | - | - |
| Linaer | $100 \times 10$ | $10$ |
| Softmax | - | $\times$[2] |

[1] "-" represents that the model has this layer with no specified configuration.

[2] "$\times$" represents that the model does not contain this layer.

Table 6: Example architectures of original AlexNet and the corresponding pill blueprint.

| Layer Type | Original AlexNet | Our Pill Blueprint |
|---|---|---|
| Input | $32 \times 32 \times 3$ | $32 \times 32 \times 3$ |
| Conv2d | $11 \times 11 \times 48$ | $11 \times 11 \times 1$ |
| ReLU | - | - |
| MaxPool2d | $3 \times 3$ | $3 \times 3$ |
| Conv2d | $3 \times 3 \times 96$ | $3 \times 3 \times 1$ |
| ReLU | - | - |
| MaxPool2d | $3 \times 3$ | $3 \times 3$ |
| Conv2d | $3 \times 3 \times 192$ | $3 \times 3 \times 1$ |
| ReLU | - | - |
| Conv2d | $3 \times 3 \times 192$ | $3 \times 3 \times 1$ |
| ReLU | - | - |
| Conv2d | $3 \times 3 \times 128$ | $3 \times 3 \times 1$ |
| ReLU | - | - |
| MaxPool2d | $3 \times 3$ | $3 \times 3$ |
| Linear | $4608 \times 1024$ | $36 \times 1$ |
| ReLU | - | - |
| Linear | $1024 \times 512$ | $1 \times 10$ |
| ReLU | - | - |
| Linear | $512 \times 10$ | $10$ |
| Softmax | - | $\times$ |

## B  DETAILED THREAT MODEL

**Attacker's Goal and Capabilities.** This paper focuses on improving the effectiveness of existing poisoning attacks in FL. Similar to previous work (Fang et al., 2020; Shejwalkar & Houmansadr, 2021), an attacker aims to raise the error rates of the global model on a specific class or multiple classes by sending poisoned model updates via compromised clients during the iterative aggregation. Our method does not require any additional knowledge compared with existing FL poisoning attacks. Hence we reuse the typical threat model in existing studies (Fang et al., 2020; Shejwalkar & Houmansadr, 2021). The attacker has a complete control of the compromised clients, including their local data, local training, and uploading process. With the aggregated resources on the compromised clients, the attacker may aggregate the local data from the compromised clients to do extra training or aggregate their local updates to do model estimation. The attacker may or may not know the updates of other benign clients, depending on the confidentiality of the communication channels between the server and clients. Besides, the attacker cannot access the server's information, including the aggregation rules or the selected clients in each round.

**Defense Settings.** Most of the defenses in FL are deployed and executed on the server. We adopt a similar defense setting as existing studies (Blanchard et al., 2017; Yin et al., 2018; Cao et al., 2021;

Guo et al., 2021). The server cannot directly analyze the local data or the local training of clients. It can only detect malicious clients through model updates from different clients. The server can collect and possess a root test dataset to provide more accurate and robust detection, while the data of such a root test dataset cannot be derived from clients. The data distribution of this root test dataset may or may not be the same as the data distribution across the clients.

## C  ADDITIONAL DETAILS OF CONCRETE PILL BLUEPRINTS

Table 5 and Table 6 illustrate the model structures of the CNN model and the simplified AlexNet, with their corresponding pill's blueprints.

## D  DETAILED PILL SEARCH ALGORITHM

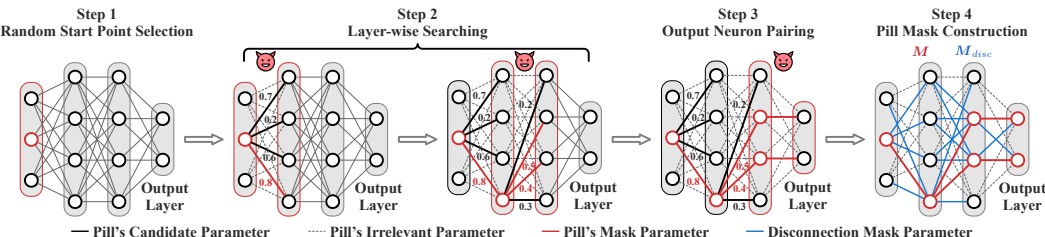

**Figure 5: An example of the "approximate max pill search" algorithm in our augmentation method.**

A complete procedure of the "approximate max pill search" algorithm consists of the following four steps. To improve readability, we use "neuron" to represent both neurons in fully connected layers and channels in convolutional layers, and we use the classification task as an example:

**Step 1 Random Start Point Selection:**  At the beginning of the search, we randomly choose a subset of neurons from the first layer $l_1$ of the target model, based on the structure of the first layer $l_1^p$ in the pill's blueprint and its neuron number $\mathcal{N}_1^p$. The selected neurons are termed as $\mathcal{V}_1$, and defined as start points, which are then fixed across the entire FL training.

**Step 2 Layer-wise Search:**  For each subsequent layer $l_i$ in the target model, we first calculate the sum of the weights from the selected neurons $\mathcal{V}_{i-1}$ in layer $l_{i-1}$ to each neuron in $l_i$. Then, we rank all the neurons in $l_i$ based on the parameter value sums and choose top $\mathcal{N}_i^p$ neurons in $l_i$ as the new $\mathcal{V}_i$, where $\mathcal{N}_i^p$ represents the number of neurons in the $i$th layer of pill's blueprint. $\mathcal{V}_i$ and all the parameters from $\mathcal{V}_{i-1}$ to $\mathcal{V}_i$ are recorded.

**Step 3 Output Neuron Pairing:**  After visiting $l_{L-1}$ layers, where $L$ is the total number of layers in the target model, $||\mathcal{V}_{L-1}||$ should equal to the neuron number in $l_L$ of the target model, which also equals to the number of classes. We select all the neurons in the target model's $l_L$ layer into our pill to construct $\mathcal{V}_L$. Then, we only record the parameters from one neuron in $\mathcal{V}_{L-1}$ to only one neuron in $\mathcal{V}_L$ based on the index order (*i.e.*, the first neuron in $\mathcal{V}_{L-1}$ is paired with the first neuron in $\mathcal{V}_L$). Since $||\mathcal{V}_{L-1}||$ equals to $||\mathcal{V}_L||$, the number of recorded parameters equals to the number of classes, avoiding poisoning too many parameters in a single layer.

**Step 4 Pill Mask Construction:**  With the recorded $\mathcal{V}_i$ and the corresponding parameters, we construct two masks $M$ and $M_{disc}$. The mask $M$ records the pill's parameters in the target model, and the disconnection mask $M_{disc}$ records the parameters of the connections between the pill and the rest of the target model. $M$ is used for poisoning, while $M_{disc}$ is used to disconnect the poison pill from the target model, maintaining the integrity and performance of the pill during poisoning. The two masks have the same shape as the target model's parameters. To construct $M$, we set the locations corresponding to the pill parameters to 1, and the others to 0. Based on $M$, we can similarly obtain the corresponding disconnected mask $M_{disc}$, which sets the locations corresponding to the parameters from neurons other than $\mathcal{V}_{i-1}$ to $\mathcal{V}_i$ in each model layer $l_i$ (except for the $L$th layer since we choose all the neurons from it), and also those corresponding to parameters from $\mathcal{V}_{i-1}$ to other pill irrelevant neurons in each model layer $l_i$ to 1. The two masks are used in the Pill Injection Stage.

**Table 7: Different dynamic patterns in our augmentation method, utilizing along with the max subnetwork searching.**

| Notation | Description |
|---|---|
| PATTERN$_1$ | All layers use the adaptive searching strategy |
| PATTERN$_2$ | All layers use the one-time searching strategy |
| PATTERN$_3$ | *FE* layers use the adaptive searching strategy
*CLS* layers use the repeated searching strategy |
| PATTERN$_4$ | *FE* layers use repeated searching strategy
*CLS* layers use the adaptive searching strategy |
| PATTERN$_5$ | *FE* layers use the adaptive searching strategy
*CLS* layers use the one-time searching strategy |
| PATTERN$_6$ | *FE* layers use the one-time searching strategy
*CLS* layers use the adaptive searching strategy |

**Example.** Figure 5 presents a concrete example of the search algorithm in a 4-layer linear model. Since the start point is randomly selected by the attacker, defense methods can hardly guess it without any prior knowledge. In the example, suppose the model is a 4-layer linear model for a binary classification task. Then the pill blue print contains one neuron in the first two layers, and contains two neurons in the last two layers. Initially, we randomly select a start neuron, specifically the second neuron in the first layer in the example. Then, we conduct layer-wise searching when visiting the second and third layers, selecting the parameters with the highest magnitudes. At the forth layer (output layer), we pair the two output neurons with the two selected neurons in the third layer based on the index order. And we finally construct two pill-related masks accordingly.

With the search algorithm, we also reduce the complexity of the pill search from $\mathcal{O}(\prod_1^L \mathcal{N}_i)$ to $\mathcal{O}(\sum_1^L \mathcal{N}_i \cdot \mathcal{N}_i^p)$, where $\mathcal{N}_i$ represents the neuron number of the target model in layer $i$, and $\mathcal{N}_i^p \ll \mathcal{N}_i$ for all the hidden layers. The computational complexity of our pill search is hence much smaller than the computational complexity of one round local training.

To make the pill search process dynamic, we also design several patterns to adaptively determine whether to change the pill in the training period, shown in Table 7 (*FE* represents the convolutional layers, *CLS* represents the linear layers). For more details about each specific dynamic pattern, please refer to Appendix E. The combination of the "approximate max pill search" with different dynamic patterns constructs the complete dynamic pill search, considering both the *stealthiness* and the *efficiency*.

## E  ADDITIONAL DETAILS OF THE DYNAMIC PATTERNS IN OUR METHOD

We first design three searching strategies, including one-time searching strategy, repeated searching strategy, and adaptive searching strategy.

In the one-time searching strategy, we search the pill based on the initial global model using the "approximate max pill search" algorithm introduced in §4.2, and keep this pill unchanged in the whole FL training. The one-time searching strategy benefits the formation of pill in the global model, while the initial pill may be less effective with the increasing training rounds of the global model, due to the changing importance of the model parameters.

On the contrary, the repeated searching strategy runs the 'approximate max pill search' algorithm in every training round. The repeated searching strategy can help our method modify more parameters in the global model, and make the pill less traceable. While the attacking effects may be reduced due to the constantly changing pill.

Considering advantages and disadvantages of both the one-time searching strategy and the repeated searching strategy, we design a more flexible searching strategy, termed as "adaptive searching strategy". In the adaptive searching strategy, our method searches the new pill only when the pill is

not successfully injected into the global model in the last round. The condition:

$$\mathtt{Sim}(M \odot \Delta g_t, M \odot \Delta g_t^{(i)}) < C_{search} \tag{3}$$

should be satisfied to trigger the new subnetwork searching on malicious client $i$, where $C_{search}$ is set as $0.94$ in the experiments. The adaptive searching strategy is a more moderate version of repeated searching.

Since the three searching strategies have their unique advantages, we investigate different combinations of them in the experiments. We further divide the neural network into *Feature Extractor (FE)* and *Classifier (CLS)*. Refer to the CNN model we used, the convolutional layers are regarded as *FE*, and the linear layers are regarded as *CLS*. We use different searching strategies in *FE* and *CLS*, respectively. In all the nine combinations, we test and keep six of them, noted as $\mathrm{PATTERN}_1$ to $\mathrm{PATTERN}_6$, shown in Table 7 in § 4.2. Such six patterns construct the entire dynamic pattern set used in our method.

# F    ADDITIONAL DETAILS OF PILL ADJUSTMENT

---

**Algorithm 2: Similarity-based and distance-based adjustment functions in the Poison Pill Injection stage.**

```
 1  function SimAdjust(param, Δg̃_{t+1}, Δg_{t+1}^{(i)})          13  function DistAdjust(param, Δg̃_{t+1}, Δg_{t+1}^{(i)})
 2      {Δg'_{t+1}^{(1),···,(m)}, M_all} ← param;                14      {Δg'_{t+1}^{(1),···,(m)}, M_all} ← param;
 3      S_max ← max(0, max{Sim(Δg̃_{t+1}, Δg_{t+1}^{(i)'}); i ∈   15      Dist_max ← max{||Δg'_{t+1}^{(i)} − Δg̃_{t+1}||; i ∈ {1,···,m}};
        {1,···,m}});                                             16      Dist ← ||Δg_{t+1}^{(i)} − Δg̃_{t+1}||;
 4      iter ← 0;                                                17      if ||C_↓ · Δg_{t+1}^{(i)} − Δg̃_{t+1}|| < ||C_↑ · Δg_{t+1}^{(i)} − Δg̃_{t+1}||
 5      while Sim(Δg̃_{t+1}, Δg_{t+1}^{(i)}) < S_max && iter <         then
        C_iter do                                                18          C_dist ← C_↓;
 6          if iter %2 then                                      19      else
 7              Δg_{t+1}^{(i)} ← (C_↑ · (1 − M_all) + M_all)     20          C_dist ← C_↑;
                ⊙ Δg_{t+1}^{(i)};                                21      while Dist ≥ Dist_max &&
 8              iter ← iter +1;                                          ||C_dist · Δg_{t+1}^{(i)} − Δg̃_{t+1}|| ≤ Dist do
 9          else                                                 22          Δg_{t+1}^{(i)} ← C_dist · Δg_{t+1}^{(i)};
10              Δg_{t+1}^{(i)} ← ((1 − M_all) + C_↓ · M_all)     23          Dist ← ||Δg_{t+1}^{(i)} − ||;
                ⊙ Δg_{t+1}^{(i)};                                24      return Δg_{t+1}^{(i)};
11              iter ← iter +1;
12      return Δg_{t+1}^{(i)};
```

---

The details of the two pill adjustment methods are presented as follows:

**Similarity-based Adjustment.** As shown in Line `1-12` of Algorithm 2, we first compute the maximum cosine similarity $S_{max}$ between the normal model updates from the compromised clients and the estimated global model update in the current round. Then, we iteratively and alternately reduce the magnitude of the poison pill's parameters with the down-scaling factor $C_{\downarrow}$, and increase the magnitude of the rest estimated global model update's parameters with the up-scaling factor $C_{\uparrow}$, until the cosine similarity between the entire poisoned model update and the estimated global model update is greater than $S_{max}$ or the adjustment total iteration is greater than the threshold $C_{iter}$.

**Distance-based Adjustment.** In the **Distance-based Adjustment** (Line `13-24` of Algorithm 2), we reuse the up-scaling factor $C_{\uparrow}$ and the down-scaling factor $C_{\downarrow}$ to adjust the magnitude of the entire poisoned model update. The intuition behind the **Distance-based Adjustment** is shown in Figure 3. We first calculate the maximum distance between the normal model updates from the compromised clients and the estimated global model update in the current round. We use this maximum distance $Dist_{max}$ as the threshold in the distance-based adjustment. Then, we further determine the scaling factor that should be used by applying the two scaling factors $C_{\uparrow}$ and $C_{\downarrow}$ separately to the poisoned model update $\Delta g_{t+1}^{(i)}$. The scaling factor that reduces the distance between $\Delta g_{t+1}^{(i)}$ and $\Delta \tilde{g}_{t+1}$ is chosen as the initial scaling factor in the subsequent iterative scaling. We stop the scaling until the distance between the $\Delta g_{t+1}^{(i)}$ and $\Delta \tilde{g}_{t+1}$ is smaller than $Dist_{max}$, or the scaling factor begins to increase such distance (reach the limit of the scaling).

## G  ADDITIONAL EXPERIMENTAL CONFIGURATIONS

**Model, Dataset, and Hyper-Parameters.** In our experiments, we employ a four-layer Convolutional Neural Network (CNN) and a simplified version of AlexNet (Krizhevsky et al., 2012). The structures of the models and their corresponding pill blueprints are detailed in Appendix C. We evaluate our method on three widely-used datasets: MNIST (LeCun, 1998), Fashion-MNIST (Xiao et al., 2017), and CIFAR-10 (Krizhevsky et al., 2009). We use the CNN model on MNIST and Fashion-MNIST datasets, and the AlexNet (Krizhevsky et al., 2012) on CIFAR-10 dataset. Each experiment is repeated five times to ensure reliability, with the mean and standard deviation (std) of the results reported.

**IID and Non-IID Data Settings.** Our method was assessed under both IID and non-IID data distributions to understand its performance across data heterogeneity. For IID data setting, we uniformly split all the training data into $K$ shards, and distribute each shard to a random client. For non-IID data setting, we utilize the non-IID degree $p$ as defined in prior studies (Fang et al., 2020; Guo et al., 2021). A higher $p$ indicates greater data heterogeneity among the clients. Specifically, when $p = 0.1$, the data configuration is essentially IID. We set $p = 0.5$ to to intensify the non-IID condition, under which we create and allocate $K$ non-IID data shards to all the clients, simulating a more realistic and challenging FL environment. Given that FLTrust necessitates a root dataset at the server, we select this dataset first from the available training data. Subsequently, we distribute the remaining data among the clients according to the aforementioned IID and non-IID rules. This approach ensures that there is no overlap between server's data and client's data.

**Configurations of Dynamic Patterns in Our Method.** As outlined in Section 4.2, we design six dynamic patterns for the pill search. We systematically evaluate all six patterns and present the results of the most effective strategy.

**Evaluation Metrics.** We use *error rates* – defined as the proportion of incorrect predictions – to evaluate attack effectiveness. Given that the model poisoning attacks discussed are all untargeted, higher error rates indicate more effective attacks. To assess the *stealthiness* of our method in delivering malicious updates, we employ two metrics: 1) *cosine similarity score*, measuring alignment with the server's model update in FLTrust; 2) *distance score*, used in Multi-Krum to evaluate the closeness of poisoned updates to benign updates.

## H  DETAILED AUGMENTATION PERFORMANCE ANALYSIS

Following are individual improvements of our method on different baseline attacks in IID data setting:

- Sign-flipping attack: Its original version achieves a high error rate due to its aggressive and brute design, but it is effective only under FedAvg. Our method extends its impact to five more defenses (Multi-Krum, Bulyan, Median, Trim, and FLD), raising the average error rate by $0.399$.

- Trim and Krum attack: Our method enables these two attacks to successfully penetrate all baseline defenses (except for Trim attack against FLD) including FLTrust, which were previously unbreachable, with average error rate increases of $0.249$ and $0.253$, respectively.

- Min-Max attack: With our method, the Min-Max Attack shows a comprehensive improvement against all defenses except for a slight decrease against Bulyan, achieving an overall average error rate increase of $0.222$.

Similarly, the detailed improvements for a specific attack in the non-IID data setting shown as follows:

- Sign-flipping attack: Our method helps the sign-flipping attack achieve an average error rate increase of $0.404$, which is similar to the error rate increase on IID data.

- Trim and Krum attack: Both attacks penetrate all baseline defenses under the enhancement of our method, with average improvements of $0.281$ and $0.236$, respectively.

- Min-Max attack: Our method helps the Min-Max attack achieve an average error rate increase of $0.195$, higher than its original version. Although this error rate increase is lower than that in the IID data setting, it remains higher than the error rate increase caused by its original version.

**Table 8: Error rates under cross-silo setting using "approximate max pill search" (20% malicious clients) on MNIST dataset.**

| Data Distribution | IID | | | | | | | Non-IID | | | | | | |
|---|---|---|---|---|---|---|---|---|---|---|---|---|---|---|
| **Attack** | **FedAvg** | **FLTrust** | **MKrum** | **Bulyan** | **Median** | **Trim** | **FLD** | **FedAvg** | **FLTrust** | **MKrum** | **Bulyan** | **Median** | **Trim** | **FLD** |
| No Attack | 0.028 | 0.051 | 0.029 | 0.029 | 0.045 | 0.029 | 0.025 | 0.029 | 0.042 | 0.030 | 0.029 | 0.041 | 0.029 | 0.022 |
| Sign-flipping Attack | **0.934** | 0.059 | 0.038 | 0.055 | 0.055 | 0.036 | 0.025 | **0.886** | **0.073** | 0.041 | 0.052 | 0.059 | 0.041 | 0.026 |
| + Poison Pill | 0.353 | **0.093** | 0.454 | 0.283 | 0.268 | **0.173** | **0.588** | 0.431 | 0.059 | **0.605** | **0.349** | **0.333** | **0.217** | **0.713** |
| Trim Attack | 0.257 | 0.065 | 0.182 | 0.103 | 0.106 | **0.123** | 0.022 | 0.418 | 0.059 | 0.295 | 0.209 | 0.245 | **0.310** | 0.021 |
| + Poison Pill | **0.416** | **0.109** | **0.469** | **0.252** | **0.247** | 0.117 | **0.026** | **0.581** | **0.065** | **0.672** | **0.358** | **0.324** | 0.092 | **0.051** |
| Krum Attack | 0.033 | 0.061 | 0.067 | 0.154 | 0.188 | 0.043 | **0.759** | 0.034 | 0.058 | 0.130 | 0.297 | 0.191 | 0.052 | **0.908** |
| + Poison Pill | **0.326** | **0.082** | **0.585** | **0.266** | **0.272** | **0.169** | 0.632 | **0.528** | **0.062** | 0.556 | **0.350** | **0.321** | **0.210** | 0.746 |
| Min-Max Attack | 0.307 | 0.082 | **0.693** | **0.731** | 0.341 | **0.255** | **0.915** | 0.359 | **0.161** | 0.718 | 0.993 | 0.381 | **0.320** | 0.853 |
| + Poison Pill | **0.402** | **0.106** | 0.518 | 0.273 | 0.262 | 0.218 | 0.766 | **0.534** | 0.077 | 0.707 | 0.369 | 0.318 | 0.194 | **0.861** |

**Table 9: Error rates under cross-device setting using "approximate max pill search" (20% malicious clients) on Fashion-MNIST dataset in both IID and non-IID data distribution.**

| Data Distribution | IID | | | | | | Non-IID | | | | | |
|---|---|---|---|---|---|---|---|---|---|---|---|---|
| **Attack** | **FedAvg** | **FLTrust** | **MKrum** | **Bulyan** | **Median** | **Trim** | **FedAvg** | **FLTrust** | **MKrum** | **Bulyan** | **Median** | **Trim** |
| No Attack | 0.107 ±0.004 | 0.111 ±0.003 | 0.108 ±0.003 | 0.105 ±0.003 | 0.138 ±0.010 | 0.106 ±0.003 | 0.113 ±0.002 | 0.124 ±0.008 | 0.115 ±0.006 | 0.118 ±0.002 | 0.164 ±0.009 | 0.116 ±0.005 |
| Sign-flipping Attack | **0.940** ±0.026 | 0.116 ±0.003 | 0.110 ±0.003 | 0.128 ±0.005 | 0.165 ±0.007 | 0.121 ±0.004 | **0.905** ±0.031 | 0.124 ±0.004 | 0.118 ±0.002 | 0.136 ±0.003 | 0.184 ±0.007 | 0.134 ±0.006 |
| + Poison Pill | 0.591 ±0.177 | **0.117** ±0.004 | **0.749** ±0.076 | **0.357** ±0.057 | **0.589** ±0.048 | **0.225** ±0.026 | 0.573 ±0.140 | **0.125** ±0.004 | **0.665** ±0.111 | **0.379** ±0.018 | **0.662** ±0.131 | **0.277** ±0.012 |
| Trim Attack | 0.240 ±0.018 | 0.110 ±0.004 | 0.151 ±0.010 | 0.148 ±0.002 | 0.207 ±0.014 | 0.178 ±0.004 | 0.340 ±0.048 | 0.120 ±0.002 | 0.228 ±0.025 | 0.190 ±0.016 | 0.237 ±0.016 | **0.245** ±**0.011** |
| + Poison Pill | **0.620** ±**0.051** | **0.492** ±**0.023** | **0.620** ±**0.025** | **0.228** ±**0.025** | **0.424** ±**0.042** | **0.232** ±**0.035** | **0.654** ±**0.037** | **0.533** ±**0.041** | **0.679** ±**0.049** | **0.324** ±**0.098** | **0.483** ±**0.098** | 0.226 ±0.025 |
| Krum Attack | 0.117 ±0.002 | 0.112 ±0.004 | 0.172 ±0.010 | 0.238 ±0.005 | 0.169 ±0.009 | 0.132 ±0.003 | 0.126 ±0.005 | 0.121 ±0.004 | 0.204 ±0.031 | 0.296 ±0.011 | 0.222 ±0.014 | 0.158 ±0.006 |
| + Poison Pill | **0.681** ±**0.057** | **0.138** ±**0.015** | **0.740** ±**0.092** | **0.362** ±**0.073** | **0.572** ±**0.167** | **0.258** ±**0.018** | **0.604** ±**0.125** | **0.141** ±**0.009** | **0.750** ±**0.081** | **0.372** ±**0.035** | **0.649** ±**0.184** | **0.277** ±**0.013** |
| Min-Max Attack | 0.146 ±0.005 | 0.111 ±0.002 | 0.382 ±0.036 | **0.324** ±**0.012** | 0.183 ±0.011 | 0.185 ±0.006 | 0.191 ±0.014 | 0.147 ±0.024 | 0.621 ±0.112 | **0.426** ±**0.072** | 0.245 ±0.013 | 0.279 ±0.007 |
| + Poison Pill | **0.651** ±**0.082** | **0.244** ±**0.104** | **0.718** ±**0.059** | 0.312 ±0.026 | **0.503** ±**0.060** | **0.249** ±**0.014** | **0.670** ±**0.123** | **0.229** ±**0.098** | **0.621** ±**0.030** | 0.349 ±0.047 | **0.581** ±**0.161** | **0.386** ±**0.141** |

## I ADDITIONAL STEALTHINESS ANALYSIS

**Distance Score Analysis.** Figure 4 compares the average distance scores of benign and malicious clients (with and without our method) across four baseline model poisoning attacks. The distance scores when using our method closely match or are even identical to those of benign clients throughout the entire training process. In contrast, original attacks like the Trim and sign-flipping attacks display distance scores that were significantly higher or lower than those of benign updates, indicating either detected by Multi-Krum (higher scores) or underutilized attack capacities (lower scores). Our method also has a lower distance score variance in the early FL training period, representing that our method provides more steady attack efficacy in the FL's critical training period (Yan et al., 2023a; 2022) by fully utilizing the attack capacities while being undetected. Additionally, our method also achieves two more improvements. First, our method causes the global model to degrade earlier compared to the original attacks, further demonstrating the effectiveness of our augmentation. Second, our method significantly increases the discrepancy between benign client updates as the communication rounds increase. While original attacks can bypass detection in some cases, the discrepancy between benign client updates remains steady, illustrating the lower impact of malicious clients. In contrast, our method consistently increases the discrepancy among benign clients, highlighting its penetrating effectiveness in its influence on benign clients' local training.

**Table 10: Error rates under cross-device setting using "approximate max pill search" (10% malicious clients) on Fashion-MNIST dataset in both IID and non-IID data distribution.**

| Data Distribution | IID | | | | | | Non-IID | | | | | |
|---|---|---|---|---|---|---|---|---|---|---|---|---|
| Attack | FedAvg | FLTrust | MKrum | Bulyan | Median | Trim | FedAvg | FLTrust | MKrum | Bulyan | Median | Trim |
| No Attack | 0.110 ±0.004 | 0.106 ±0.004 | 0.107 ±0.005 | 0.108 ±0.003 | 0.139 ±0.008 | 0.106 ±0.003 | 0.115 ±0.003 | 0.117 ±0.003 | 0.115 ±0.003 | 0.117 ±0.002 | 0.164 ±0.004 | 0.112 ±0.002 |
| Sign-flipping Attack | **0.929** ±**0.026** | 0.111 ±0.004 | 0.108 ±0.003 | 0.111 ±0.002 | 0.153 ±0.025 | 0.117 ±0.004 | **0.902** ±**0.034** | 0.118 ±0.001 | 0.115 ±0.004 | 0.120 ±0.003 | 0.175 ±0.008 | 0.134 ±0.008 |
| + Poison Pill | 0.195 ±0.032 | **0.114** ±**0.003** | **0.170** ±**0.071** | **0.138** ±**0.005** | **0.347** ±**0.059** | **0.137** ±**0.004** | 0.330 ±0.135 | **0.124** ±**0.006** | **0.165** ±**0.016** | **0.148** ±**0.007** | **0.483** ±**0.225** | **0.161** ±**0.009** |
| Trim Attack | 0.112 ±0.003 | 0.114 ±0.004 | 0.111 ±0.002 | 0.118 ±0.005 | 0.153 ±0.021 | 0.113 ±0.004 | 0.129 ±0.003 | 0.125 ±0.004 | 0.128 ±0.004 | 0.129 ±0.003 | 0.185 ±0.011 | 0.122 ±0.003 |
| + Poison Pill | **0.369** ±**0.147** | **0.138** ±**0.015** | **0.212** ±**0.066** | **0.128** ±**0.005** | **0.310** ±**0.038** | **0.140** ±**0.014** | **0.589** ±**0.046** | **0.154** ±**0.017** | **0.300** ±**0.100** | **0.139** ±**0.006** | **0.351** ±**0.056** | **0.156** ±**0.015** |
| Krum Attack | 0.110 ±0.003 | 0.113 ±0.001 | 0.115 ±0.003 | 0.128 ±0.004 | 0.144 ±0.004 | 0.113 ±0.003 | 0.121 ±0.002 | 0.116 ±0.001 | 0.123 ±0.004 | 0.135 ±0.004 | 0.183 ±0.008 | 0.120 ±0.001 |
| + Poison Pill | **0.164** ±**0.038** | **0.117** ±**0.003** | **0.157** ±**0.044** | **0.143** ±**0.010** | **0.371** ±**0.034** | **0.142** ±**0.005** | **0.229** ±**0.069** | **0.126** ±**0.002** | **0.249** ±**0.167** | **0.146** ±**0.004** | **0.374** ±**0.023** | **0.157** ±**0.005** |
| Min-Max Attack | 0.116 ±0.002 | 0.111 ±0.002 | 0.116 ±0.001 | 0.127 ±0.004 | 0.145 ±0.006 | 0.122 ±0.003 | 0.121 ±0.002 | 0.116 ±0.001 | 0.123 ±0.004 | 0.135 ±0.004 | 0.183 ±0.008 | 0.120 ±0.001 |
| + Poison Pill | **0.351** ±**0.204** | **0.124** ±**0.019** | **0.299** ±**0.110** | **0.135** ±**0.004** | **0.343** ±**0.070** | **0.146** ±**0.019** | **0.342** ±**0.076** | **0.138** ±**0.018** | **0.292** ±**0.087** | **0.148** ±**0.009** | **0.417** ±**0.050** | **0.166** ±**0.010** |

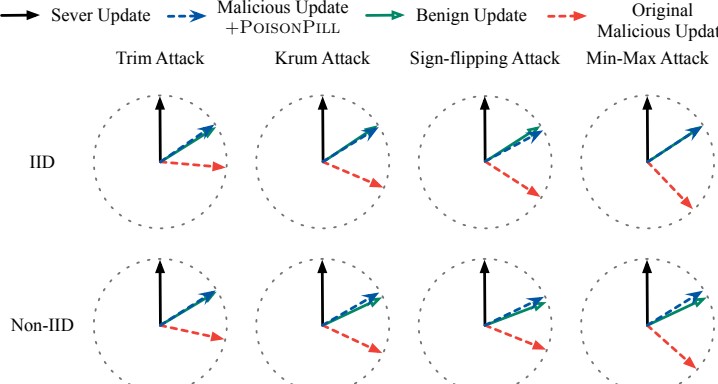

**Figure 6: Comparison of cosine similarity scores between original attack with and without our method under FLTrust.**

**Cosine Similarity Score Analysis.** Figure 6 shows that the angles between server model updates and malicious updates using our method are similar or even smaller than those of benign updates, leading to higher aggregation weights for malicious updates in FLTrust – illustrating why our method makes existing FL poisoning attacks effectively bypass FLTrust. In contrast, the angles between the FLTrust's server model updates with original malicious updates are often greater than 90°, leading to a zero aggregation weight. Detailed per-round cosine similarity trends (Figure 7) also reveal that while original attacks often result in negative similarities (and thus are excluded by FLTrust), our method maintains positive similarities throughout the entire training process. This consistency not only ensures the successful insertion of the pill in any specific round but also secures pill's long-lasting presence in the global model.

## J ADDITIONAL DETAILS ON MNIST AND CIFAR-10 DATASET

The detailed results on MNIST and CIFAR-10 datasets are presented in Table 8 (MNIST dataset) and Table 4 (CIFAR-10 dataset), respectively.

For MNIST dataset, the highest error rate increase achieved using our method is 0.518, with an average increase of 0.121. This average error rate increase is slightly lower compared with the

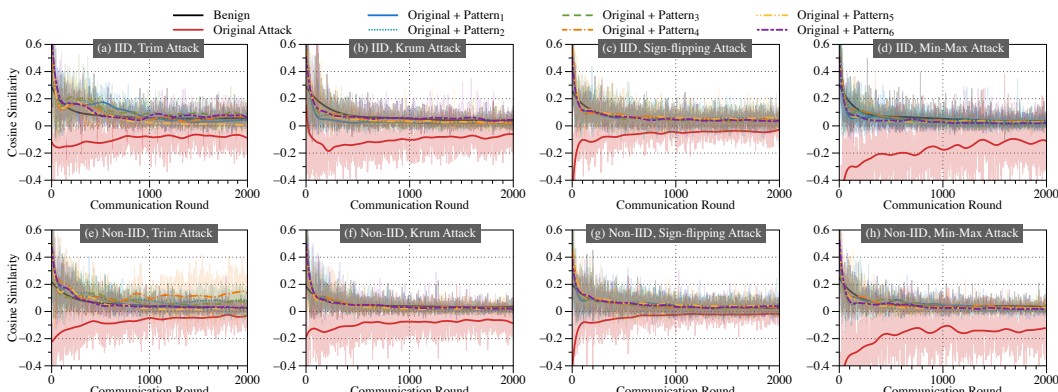

**Figure 7:** Comparison of the cosine similarity scores with the server model of the original attack and our different augmentation patterns in the entire training period under FLTrust.

improvement observed on the Fashion-MNIST dataset. Despite the reduced average error rate increase, it remains significant, especially considering the MNIST dataset's lower baseline error rates (below 0.070).

On CIFAR-10 dataset, our method helps existing FL poisoning attacks outperform their original versions in 71 of the 72 scenarios, with an average error rate increase of over 0.288. Specifically, our method facilitates at least a 0.212 increase in error rates against FLTrust, outperforming the results in the same settings on the Fashion-MNIST dataset.

## K ADDITIONAL RESULTS IN CROSS-DEVICE FL SYSTEM

After evaluating our method in the 50-client cross-silo FL system, we further test it in the 50-client cross-device FL system. Table 9 presents the error rates under the cross-device FL setting using the "approximate max pill search" algorithm on both IID and non-IID data. We report the highest error rates among the results of six dynamic patterns, with the malicious client proportion set to 20%. Since FLD is not typically designed for cross-device systems, we do not test it in this setting.

**Results on IID Data.** The highest error rate improvement with our method achieves 0.639, and the average error rate increase with our method reaches 0.279. With our method, existing model poisoning attacks outperform their original versions in 22 out of the 24 cases. The highest error rate improvement for the sign-flipping attack is 0.639, with an average error rate increase of 0.279. For the Trim attack and Krum attack, the highest error rate increases are 0.469 and 0.568, with average error rate increases of 0.264 and 0.302, respectively. For the Min-Max attack, the highest error rate increase reaches 0.505, with an average increase of 0.272. These improvements are consistent with the error rates observed under the cross-silo FL setting using the "approximate max pill search" algorithm on IID data.

**Results on non-IID Data.** As for the results on non-IID data, the highest error rate improvement with our method achieves 0.546, and the average error rate increase with our method reaches 0.273. By using our method, existing model poisoning attacks outperform their original versions in 21 out of the 24 cases. The highest error rate improvement for the sign-flipping attack is 0.547, with an average error rate increase of 0.282. For the Trim attack and Krum attack, the highest error rate rises are 0.451 and 0.546, with an average error rate rise of 0.312 and 0.278, respectively. For the Min-Max attack, the highest error rate increase reaches 0.479, with an average increase of 0.201. These improvements are also aligned with the error rates observed under the cross-silo FL setting using the max subnetwork searching algorithm on non-IID data.

The average error rates of the global model in the cross-device FL system are lower than the error rates in the cross-silo FL system within 0.030, illustrating our method's generality over different data distribution and FL systems.

## L ADDITIONAL RESULTS WITH FEWER MALICIOUS CLIENTS

We also test the error rate improvement of our method in both the IID and non-IID cross-device FL systems, with only $10\%$ malicious clients. The experimental results are shown in Table 10.

**Results on IID Data with Fewer Malicious Clients.** The highest error rate increment is $0.257$, with an average increment of $0.083$. The error rate increments in the cross-device FL system are smaller than those in the cross-silo FL system, as malicious clients may not be selected in every round. However, this reduction in improvement is acceptable since our method helps existing model poisoning attacks outperform their original versions in 23 out of 24 cases. Furthermore, when all existing attacks fail to bypass any defenses with $10\%$ malicious clients, our method enables the attacks to bypass all defenses. The superiority of our method is maintained even with $10\%$ compromised clients.

**Results on Non-IID Data with Fewer Malicious Clients.** The results on the non-IID data are similar to those on the IID data. The highest error rate increment is $0.460$, and the average error rate increment is $0.079$. Our method helps existing model poisoning attacks achieve higher error rates in 23 out of 24 cases, even in highly unstable and heterogeneous settings. These results demonstrate the generality and robustness of our method across different data distributions and client selection methods with only a small portion of malicious clients.

## M IMPACT OF THE PILL SEARCH ALGORITHM

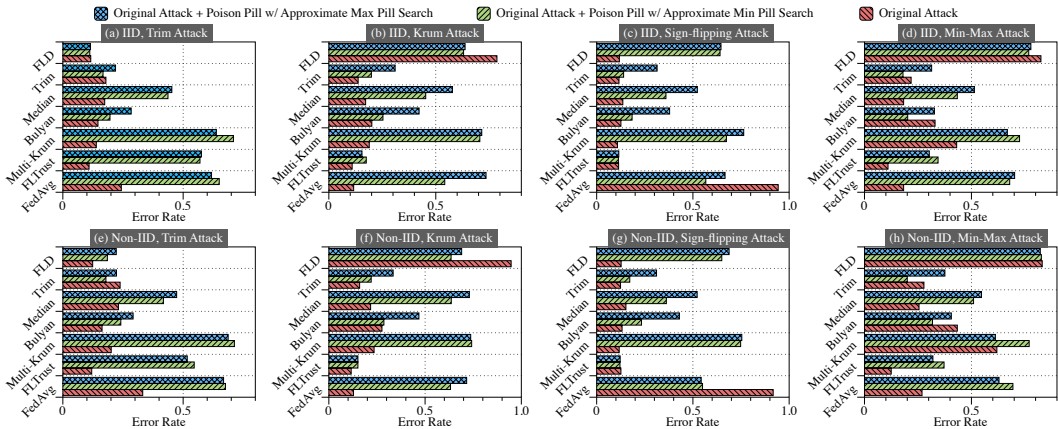

**Figure 8: Comparison of error rates between original poisoning attacks and our method with two different pill search methods.**

We conduct a final evaluation to assess the importance and effectiveness of the "approximate max pill search" algorithm used in our method. This is contrasted against a newly devised "approximate min pill search" algorithm, which targets the least important parameters within the target model. Figure 8 illustrates the error rates achieved by the "approximate max pill search", the "approximate min pill search", and the original model poisoning attacks. The "approximate max pill search" algorithm outperforms the "approximate min pill search" in 41 out of 56 cases (approximately 73%), underscoring its effectiveness in leveraging the most influential parameters to enhance attack impacts. Despite its lower efficacy, the "approximate min pill search" still manages to surpass the original attacks in 41 out of 56 cases (approximately 73%). This demonstrates the generality of our method across different pill search algorithms.

## N OUR METHOD AGAINST POSSIBLE ADAPTIVE DEFENSE

We develop an adaptive defense named DSTrust, which enhances the FLTrust's mechanism. DSTrust incorporates both distance and cosine similarity scores into a unified trust score calculation, directly countering our method's two-step adjustment approach. The round-$t$ trust score of client $i$ in DSTrust

**Table 11: Error rates under cross-silo setting against the new adaptive defense – DSTrust – with and without our method on Fashion-MNIST dataset (20% malicious clients).**

| Data Distribution | IID | | Non-IID | |
|---|---|---|---|---|
| **Attack** | w/o Poison Pill | w/ Poison Pill | w/o Poison Pill | w/ Poison Pill |
| No Attack | 0.108 | | 0.116 | |
| Sign-flipping Attack | 0.111 | **0.129** | 0.110 | **0.131** |
| Trim Attack | 0.109 | **0.629** | 0.115 | **0.630** |
| Krum Attack | 0.111 | **0.140** | 0.120 | **0.128** |
| Min-Max Attack | 0.127 | **0.167** | 0.143 | **0.327** |

is calculated as follows:

$$TS_i = ReLU\left(\frac{cos(\Delta \boldsymbol{g}_t^{(i)}, \Delta \boldsymbol{g}_t^s)}{||\Delta \boldsymbol{g}_t^{(i)} - \Delta \boldsymbol{g}_t^s||}\right), \tag{4}$$

where $\Delta \boldsymbol{g}_t^{(i)}$ represents the model update from client $i$ and $\Delta \boldsymbol{g}_t^s$) represents server's model update. By integrating both cosine similarity and distance metrics, DSTrust provides a more comprehensive defense approach compared with FLTrust. This dual consideration allows DSTrust to effectively mitigate attacks that manipulate either of these metrics to bypass defenses.

Table 11 details the error rates for four baseline FL poisoning attacks both with and without our method against the DSTrust defense on the Fashion-MNIST dataset within a 50-client FL system, where 20% of clients are malicious. These tests were conducted under both IID and non-IID data environments. DSTrust effectively neutralizes the four baseline poisoning attacks when our method is not applied, highlighting its robustness as a defense mechanism. Despite DSTrust's integration of both cosine similarity and distance metrics in its defense strategy, it fails to counteract the augmented attacks when our method is employed. Notably, our method achieves a maximum error rate increase of $0.521$, and an average error rate increase of $0.173$ across all $8$ test scenarios. These results demonstrate that merely understanding the adjustment strategies of our method, and subsequently integrating corresponding defense metrics, does not fundamentally negate the effectiveness of our method. Despite the adaptive defense's attempt to incorporate both cosine similarity and distance metrics into DSTrust, it remains insufficient to thwart the enhanced capabilities of our method.

## O    LIMITATIONS AND FUTURE WORK

Our method significantly enhances non-state-of-the-art (non-SOTA) model poisoning attacks, enabling them to SOTA results against various prevalent defenses. This is accomplished through a pill-based, attack-agnostic augmentation pipeline. We not only demonstrate our method's capabilities but also expose fundamental vulnerabilities within the current designs of defense mechanisms.

For future attacks in FL, it is essential for attackers to meticulously evaluate the importance of each parameter in their implementation. By targeting specific subsets of parameters, attackers can devise more flexible and adaptive attacks, improving *stealthiness* and complicating defense efforts. As for future defenses, while individually checking each parameter might seem viable, its practical deployment is hindered by high overheads, making it infeasible in real-world applications.

Thus, there is a pressing need for more sophisticated defenses that can conduct fine-grained analyses of the roles of different parameters in neural networks, while executing without imposing prohibitive computational costs.

## P    ADDITIONAL ABLATION STUDY ON PILL ADJUSTMENT

To illustrate the necessity of both the `SimAdjust()` and `DistAdjust()` used in our method, we conduct a detailed ablation study, providing the error rates of the Trim Attack with different settings

**Table 12: Ablation study on the necessity of both SimAdjust and DistAdjust on Fashion-MNIST dataset with 20% malicious clients.**

| Data Distribution | IID | | | | | | |
|---|---|---|---|---|---|---|---|
| Attack | FedAvg | FLTrust | MKrum | Bulyan | Median | Trim | FLD |
| Trim Attack | 0.243 | 0.109 | 0.139 | 0.146 | 0.174 | 0.179 | 0.116 |
| + Poison Pill w/ SimAdjust w/ DistAdjust | **0.618** | **0.576** | **0.638** | **0.284** | **0.453** | 0.219 | 0.115 |
| + Poison Pill w/o SimAdjust w/ DistAdjust | 0.317 | 0.105 | 0.364 | 0.247 | 0.368 | 0.136 | **0.208** |
| + Poison Pill w/ SimAdjust w/o DistAdjust | 0.554 | 0.104 | 0.122 | 0.108 | 0.429 | **0.284** | 0.119 |

**Table 13: Error rates under cross-silo setting using "approximate max pill search" (20% malicious clients) on IID Fashion-MNIST dataset with label-flipping attack.**

| Data Distribution | IID | | | | | | |
|---|---|---|---|---|---|---|---|
| Attack | FedAvg | FLTrust | MKrum | Bulyan | Median | Trim | FLD |
| No Attack | 0.109 | **0.107** | 0.105 | 0.105 | 0.123 | 0.106 | 0.115 |
| Label-flipping Attack | **0.960** | 0.107 | 0.095 | 0.105 | 0.116 | 0.115 | 0.096 |
| + Poison Pill | 0.255 | 0.105 | **0.171** | **0.231** | **0.827** | **0.406** | **0.962** |

of pill adjustment on the IID Fashion-MNIST dataset within a 50-client FL system in Table 12. The results demonstrate that simultaneously using both SimAdjust and DistAdjust outperforms using only SimAdjust or DistAdjust in 5 out of 7 cases. While using just one adjustment method may surpass the combined approach in one or two specific scenarios, it does not ensure consistent bypassing effectiveness across diverse defenses. This highlights the necessity of the combined adjustment in our method, which leverages the complementary strengths of both SimAdjust and DistAdjust to achieve superior performance.

## Q    ADDITIONAL RESULTS WITH LABEL-FLIPPING ATTACK

In addition to the untargeted model poisoning attacks discussed in the main text, we evaluate our method using a data-poisoning-based targeted attack: the label-flipping attack. Label-flipping is a straightforward yet effective targeted attack in federated learning (FL) and is also among the least stealthy data-poisoning-based attacks. To make the evaluation more challenging, we configured the attacker to flip all the labels of the training data on malicious clients, making the label-flipping attack even less stealthy. The results are shown in Table 13, demonstrating that our method enhances the label-flipping attack to bypass five additional defenses compared to its original version. This illustrates the compatibility of our method with data-poisoning-based and targeted attacks.

## R    ADDITIONAL RESULTS WITH MORE COMPLEX MODELS

To further demonstrate the effectiveness of our method on more complex model architectures, we test our method using VGG-11 net on the IID CIFAR-10 dataset, shown in Table 14. The results demonstrate that our method consistently enhances the performance of both the Trim Attack and the sign-flipping attack, outperforming their original versions in 14 out of 18 cases. These findings illustrate the effectiveness of our method with more complex model architectures.

**Table 14: Error rates under cross-silo setting using "approximate max pill search" (20% malicious clients) on IID CIFAR-10 dataset with VGG-11 net.**

| Data Distribution | IID | | | | | | | | |
|---|---|---|---|---|---|---|---|---|---|
| Attack | FedAvg | FLTrust | MKrum | Bulyan | Median | Trim | DnC | FLD | Flame |
| No Attack | 0.319 | 0.328 | 0.338 | 0.324 | 0.330 | 0.337 | 0.315 | 0.334 | 0.336 |
| Sign-flipping Attack | **0.897** | 0.335 | 0.336 | 0.329 | 0.353 | 0.386 | 0.341 | 0.316 | 0.367 |
| + Poison Pill | 0.711 | **0.483** | **0.503** | **0.457** | **0.385** | **0.410** | **0.413** | **0.898** | **0.487** |
| Trim Attack | 0.431 | 0.323 | 0.422 | 0.428 | **0.434** | **0.432** | 0.340 | **0.339** | 0.362 |
| + Poison Pill | **0.490** | **0.578** | **0.595** | **0.506** | 0.428 | 0.392 | **0.406** | 0.295 | **0.383** |

**Table 15: Error rates under cross-silo setting using "approximate max pill search" (20% malicious clients) on IID Fashion-MNIST dataset within a 100-client FL system.**

| Data Distribution | IID | | | | | | |
|---|---|---|---|---|---|---|---|
| Attack | FedAvg | FLTrust | MKrum | Bulyan | Median | Trim | FLD |
| No Attack | 0.093 | 0.097 | 0.093 | 0.094 | 0.111 | 0.092 | 0.093 |
| Trim Attack | 0.274 | **0.126** | 0.108 | 0.105 | 0.191 | **0.219** | 0.097 |
| + Poison Pill | **0.336** | 0.101 | **0.901** | **0.281** | **0.272** | 0.122 | **0.518** |

## S  ADDITIONAL RESULTS WITH LARGER FL SYSTEMS

To further demonstrate the effectiveness of our method in larger FL systems, we extend our experiments on the Fashion-MNSIT dataset with 100 clients, shown in Table 15. The results show a similar trend as observed in the 50-client system. Our method enables baseline attacks to successfully bypass four additional defenses, causing over 50% additional error rates in the global model. These findings further validate the effectiveness and generality of our approach in larger systems, when a single malicious client has fewer data samples.

## T  COMPARISON WITH EXISTING ATTACK ENHANCEMENT METHOD

**Table 16: Comparison with Neurotoxin under cross-silo setting on IID Fashion-MNIST dataset within a 50-client FL system.**

| Data Distribution | IID | | | | | | |
|---|---|---|---|---|---|---|---|
| Attack | FedAvg | FLTrust | MKrum | Bulyan | Median | Trim | FLD |
| No Attack | 0.109 | 0.107 | 0.105 | 0.105 | 0.123 | 0.106 | 0.115 |
| Sign-flipping Attack | **0.943** | 0.114 | 0.108 | 0.126 | 0.136 | 0.116 | 0.118 |
| + Neurotoxin | 0.710 | **0.147** | 0.105 | 0.103 | 0.106 | 0.105 | 0.110 |
| + Poison Pill | 0.667 | 0.115 | **0.764** | **0.379** | **0.523** | **0.314** | **0.646** |
| Trim Attack | 0.243 | 0.109 | 0.139 | 0.146 | 0.174 | 0.179 | **0.116** |
| + Neurotoxin | 0.135 | 0.109 | 0.113 | 0.106 | 0.126 | 0.119 | 0.108 |
| + Poison Pill | **0.618** | **0.576** | **0.638** | **0.284** | **0.453** | **0.219** | 0.115 |

Considering several prior studies (Bagdasaryan et al., 2020; Bhagoji et al., 2019; Zhang et al., 2022b) enhancing backdoor attacks, we also adapt one recent one - Neurotoxin (Zhang et al., 2022b) - to our untargeted attacks evaluation setting. Table 16 illustrates the results on the IID Fashion-MNIST dataset within a 50-client FL system using Trim Attack. The results demonstrate that our method outperforms Neurotoxin in 12 out of 14 cases. This highlights that directly transferring existing

**Table 17:** **Performance when the number of malicious clients is gradually decreasing.**

| Data Distribution | IID | | | | | | |
|---|---|---|---|---|---|---|---|
| Attack | FedAvg | FLTrust | MKrum | Bulyan | Median | Trim | FLD |
| No Attack | 0.109 | 0.107 | 0.105 | 0.105 | 0.123 | 0.106 | 0.115 |
| Sign-flipping Attack | **0.935** | **0.123** | 0.098 | 0.101 | 0.106 | 0.101 | 0.103 |
| + Poison Pill | 0.153 | 0.104 | **0.216** | **0.195** | **0.251** | **0.146** | **0.584** |
| Trim Attack | 0.102 | **0.112** | 0.101 | 0.106 | 0.113 | 0.103 | 0.095 |
| + Poison Pill | **0.206** | 0.102 | **0.285** | **0.163** | **0.314** | **0.109** | **0.300** |

methods designed for backdoor attacks may not yield consistent effectiveness when applied to untargeted attack scenarios. The results further validate the robustness of our approach.

## U    RESULTS WITH DECREASING NUMBER OF MALICIOUS CLIENTS

To further demonstrate the effectiveness of our method in a more practical setting, we evaluate its performance as the number of malicious clients in the FL system gradually decreases. Specifically, we used the Fashion-MNIST dataset within a 50-client FL system. Initially, 20% of clients are malicious, and for every T/4 rounds (where T is the total number of FL communication rounds), the proportion of malicious clients reduces by 5%. Here is a detailed breakdown of this setup:

- $0 \rightarrow \frac{T}{4}$: 20% clients in the FL system are malicious.
- $\frac{T}{4} \rightarrow \frac{T}{2}$: 15% clients in the FL system are malicious.
- $\frac{T}{2} \rightarrow \frac{3T}{4}$: 10% clients in the FL system are malicious.
- $\frac{3T}{4} \rightarrow T$: 5% clients in the FL system are malicious.

The results are presented in Table 17, demonstrating that our method significantly enhances the error rates achieved by the original Trim Attack and sign-flipping attack in 11 out of 14 cases, with an average error rate increase of over 50%. These findings illustrate the robustness and effectiveness of our method in a more practical scenario.

