# OpenReview forum: "Poisoning with A Pill: Circumventing Detection in Federated Learning"
_ICLR.cc/2025/Conference — Submitted to ICLR 2025_

### Official Review · Reviewer_rLSw · 2024-10-25

**Soundness:** 3
**Presentation:** 3
**Contribution:** 3
**Rating:** 8
**Confidence:** 3

**Summary:**

This paper is about model poisoning attacks by augmenting current poisoning attacks with efficient parameter update methodology. The author's critical observation is that not all model parameters contribute equally during inference. Therefore, if they can selectively modify the subset of parameters (critical parameters of pill) contributing, they will be able to make the attack more stealthy and ensure that it evades current defenses. They poison the pill using existing FL poisoning attacks, and carefully inject the poison pill into the target model. To raise suspicion, they ensure that the magnitude of parameter change is balanced with benign parameter's model update. They evaluated using 9 different aggregation rules, 4 model poisoning models and used well-known MNIST, Fashion-MNIST, and CIFAR-10 datasets to validate their claim. They also evaluated using both IID and non-IID data.

**Strengths:**

1. Their method can incorporate both cross-silo and cross-device configurations.
2. Their method is general with respect to the poisoning attack they are augmenting.
3. Using their augmentation method, they can make non-SOTA poisoning attacks perform as well as SOTA model poisoning attacks.
4. In the appendix, they show that even 10% of compromised clients can evade detection when the other attacks fail.
5. Even with non-IID data, they outperform the existing attacks in 23 out of 24 cases and achieve high error rates even in unstable and heterogeneous environments.
6. They even provided a defense against their method. It would be nice to have some evaluation regarding it.

**Weaknesses:**

1. Some details regarding the parameter update are missing, such as the frequency of the model updates and how they are scheduled.  Do the compromised and non-compromised models send their updates simultaneously? Also, do they send their updates at regular intervals or differently?
2. The parameter estimation process needs to be explained more in-depth concerning the other eight aggregation rules used in the paper. Can you give me an example of how it would work for Median, FLTrust, and Flame?
3. The threat model is somewhat weak since it assumes they completely control the compromised clients. Can you tell me how the error rate would change under this scenario, where, let's say, we start with 20% of the fleet as compromised, then after some time, the number of compromised fleets decreases to 15%, then 10%, and finally 5%?

**Questions:**

1. How frequently do the compromised and benign clients update their models? Can the compromised clients send their update after some iterations where no poisoned update has been sent?
2. In a scenario where the current benign model updates that will be sent are severely out of proportion compared to their old updates, will the attack be detected? Since the compromised client sent their model update, calibrated according to the magnitude of previous benign model updates?
3. How does the estimation process work for different aggregation methods used in federated learning? Is it always the mean of all the compromised model weights? This estimation procedure makes sense when the FedAvg aggregation rule is used, but it also seems to work for Median, FLTrust, and Flame. Can you prove your perspective on this?
4. In the pill search algorithm, how important is the initialization? What is the effect of different types of initialization?

---

> ### Author Response · Authors · 2024-11-28
> **Response to Reviewer rLSw (Part 1)**
>
> Thanks for your appreciation and suggestions. Here are our point-by-point responses:
>
> > W1: Some details regarding the parameter update are missing, such as the frequency of the model updates and how they are scheduled. Do the compromised and non-compromised models send their updates simultaneously? Also, do they send their updates at regular intervals or differently?
>
> In our experiments, we follow the commonly used settings from prior studies [1, 2, 3], which assume synchronous Federated Learning (FL) systems.
>
> At each iteration in a synchronous FL system, a selected subset of clients participates. In cross-silo settings, all clients are selected, while in cross-device settings, 40% of the clients are randomly chosen for each iteration. These selected clients receive the global model from the server and perform local training or poisoning. From the server’s perspective, both benign and malicious clients are treated equally during this process.
>
> After completing their local training or poisoning, the selected clients upload their local updates to the server. Consistent with prior studies [1, 2, 3], we do not enforce simultaneous uploads from all clients. This design reflects practical scenarios where clients may have varying computational resources, and the network bandwidth between the server and clients may differ. Once all updates from the selected clients are received, the server begins its detection and aggregation processes.
>
> This setup reflects real-world FL deployments and ensures that our experiments align with prior studies [1, 2, 3] for comparability and practicality.
>
> ---
>
> > W2: The parameter estimation process needs to be explained more in-depth concerning the other eight aggregation rules used in the paper. Can you give me an example of how it would work for Median, FLTrust, and Flame?
>
> The parameter estimation process in our method is independent of the specific aggregation rules (e.g., Median, FLTrust, or Flame). It calculates the average of the benignly trained model updates from malicious clients. The primary objective is to ensure that the estimated model is not biased toward the data of any single compromised client.
>
> This design aligns with the threat model used in this paper where the attacker does not have knowledge about the aggregation rules used by the server. It also ensures that the aggregation-based estimation is closer to the actual global model update than any individual benignly trained model update from a specific compromised client.

---

> ### Author Response · Authors · 2024-11-28
> **Response to Reviewer rLSw (Part 2)**
>
> > W3: The threat model is somewhat weak since it assumes they completely control the compromised clients. Can you tell me how the error rate would change under this scenario, where, let's say, we start with 20% of the fleet as compromised, then after some time, the number of compromised fleets decreases to 15%, then 10%, and finally 5%?
>
> Thank you for your insightful review regarding the threat model. The threat model in our paper aligns with those used in prior studies [1,2,3], where the attacker has complete control over the compromised clients. Additionally, to comprehensively evaluate the effectiveness of our method, we presented results for scenarios with 20% (Table 2) and 10% (Table 3) malicious clients in the FL system in the main text.
>
> To address your concern regarding a more realistic scenario, we conducted experiments following your proposed setting. Specifically, we used the Fashion-MNIST dataset within a 50-client FL system. Initially, 20% of clients are malicious, and for every T/4 rounds (where T is the total number of FL communication rounds), the proportion of malicious clients reduces by 5%. Below is a detailed breakdown of this setup:
> |        FL Rounds       | 0 - T/4 | T/4 - T/2 | T/2 - 3T/4 | 3T/4 - T |
> |:----------------------:|:-------:|:---------:|:----------:|:--------:|
> | Mal. Client Proportion |   20%   |    15%    |     10%    |    5%    |
>
> This setup should align with your proposed scenario. Below, we present the results using Trim Attack as the baseline attack:
> |     Attack     |   FedAvg  |  FLTrust  |   MKrum   |   Bulyan  |   Median  |    Trim   |    FLD    |
> |:--------------:|:---------:|:---------:|:---------:|:---------:|:---------:|:---------:|:---------:|
> |    No Attack   |   0.109   |   0.107   |   0.105   |   0.105   |   0.123   |   0.106   |   0.115   |
> |   Trim Attack  |   0.102   | **0.112** |   0.101   |   0.106   |   0.113   |   0.103   |   0.095   |
> | w/ Poison Pill | **0.206** |   0.102   | **0.285** | **0.163** | **0.314** | **0.109** | **0.300** |
>
> Below are the results using the Sign-flipping Attack as the baseline attack:
> |        Attack        |   FedAvg  |  FLTrust  |   MKrum   |   Bulyan  |   Median  |    Trim   |    FLD    |
> |:--------------------:|:---------:|:---------:|:---------:|:---------:|:---------:|:---------:|:---------:|
> |       No Attack      |   0.109   |   0.107   |   0.105   |   0.105   |   0.123   |   0.106   |   0.115   |
> | Sign-flipping Attack | **0.935** | **0.123** |   0.098   |   0.101   |   0.106   |   0.101   |   0.103   |
> |    w/ Poison Pill    |   0.153   |   0.104   | **0.216** | **0.195** | **0.251** | **0.146** | **0.584** |
>
> The results demonstrate that our method significantly enhances the error rates achieved by the original Trim Attack and Sign-flipping Attack in 11 out of 14 cases, with an average error rate increase of over 50%. These findings illustrate the robustness and effectiveness of our method in a more practical scenario.
>
> ---
>
> > Q1: How frequently do the compromised and benign clients update their models? Can the compromised clients send their update after some iterations where no poisoned update has been sent?
>
> Following the settings used in prior studies [1, 2, 3], we evaluate our method in both cross-silo and cross-device FL settings:
> 1. Cross-Silo Setting: All clients, including both benign and malicious ones, participate in each round of FL training. This means all clients upload their local updates in every iteration.
> 2. Cross-Device Setting: In each round, only 40% of the clients are randomly selected to participate in FL training. As a result, the actual proportion of malicious clients among the selected 40% varies across rounds. It is possible that no malicious clients are selected in some rounds, meaning no poisoned updates are sent during those rounds.
>
> This variability in the cross-device setting reflects practical FL scenarios, where client participation in training can fluctuate due to random selection.
>
> ---
>
> > Q2: In a scenario where the current benign model updates that will be sent are severely out of proportion compared to their old updates, will the attack be detected? Since the compromised client sent their model update, calibrated according to the magnitude of previous benign model updates?
>
> The estimated model in our method is the average of the benignly trained model updates from malicious clients in the current round, not from the previous round. As a result, the reference model adapts to the current behavior of benign models, ensuring it remains aligned with their updates. This design prevents the issue of mismatched proportions between old and current updates, as the estimated model dynamically reflects the latest benign update behavior.

---

> ### Author Response · Authors · 2024-11-28
> **Response to Reviewer rLSw (Part 3)**
>
> > Q3: How does the estimation process work for different aggregation methods used in federated learning? Is it always the mean of all the compromised model weights? This estimation procedure makes sense when the FedAvg aggregation rule is used, but it also seems to work for Median, FLTrust, and Flame. Can you prove your perspective on this?
>
> As mentioned in the response to W2, the estimation process in our method is independent of the specific aggregation rules (e.g., Median, FLTrust, or Flame). It calculates the average of the benignly trained model updates from malicious clients. The primary objective is to ensure that the estimated model is not biased toward the data of any single compromised client.
>
> We acknowledge that this estimation procedure might not yield the optimal parameters for aggregation rules like Median or Krum, which are not averaging-based. However, under the threat model assumed in this paper, the attacker does not have knowledge of the specific aggregation rule used by the server. Therefore, the safest and most practical approach is to estimate the update using the most widely recognized and frequently used method, which is averaging.
>
> This design ensures general applicability and avoids introducing assumptions about the server's aggregation method that could reduce the attack's robustness.
>
> ---
>
> > Q4: In the pill search algorithm, how important is the initialization? What is the effect of different types of initialization?
>
> We are uncertain about the exact meaning of "initialization" in this question. From our point of view, it could refer to the parameter initialization of the global model or the start point in the pill search in our method.
>
> If the term refers to the initialization of model parameters, we think this has minimal impact on our method. The pill poisoning process operates on model updates rather than the model weights themselves. Additionally, our method uses a reference model to ensure that the generated updates remain close to benign updates. This design mitigates any potential impact of initialization differences, ensuring consistency and robustness in the pill poisoning process.
>
> If "initialization" refers to the starting point in the pill search algorithm, it does not influence the effectiveness of our method. Regardless of the starting point, the pill search algorithm will identify a corresponding effective pill. If the attacker has no specific requirement, the starting point could be randomly initialized to further reduce the trackability of the pill.
>
> ---
>
> **References**
>
> 1. Shejwalkar, Virat, and Amir Houmansadr. "Manipulating the byzantine: Optimizing model poisoning attacks and defenses for federated learning." Network and Distributed System Security Symposium (NDSS). 2021.
> 2. Fang, Minghong, et al. "Local model poisoning attacks to {Byzantine-Robust} federated learning." 29th USENIX Security Symposium (USENIX Security 20). 2020.
> 3. Cao, Xiaoyu, et al. "FLTrust: Byzantine-robust Federated Learning via Trust Bootstrapping." Network and Distributed System Security Symposium (NDSS). 2021.

---

> ### Author Response · Authors · 2024-12-02
> **Kind Reminder from Authors**
>
> Dear Reviewer rLSw,
>
> We sincerely appreciate your valuable suggestions, which have significantly enhanced the quality of our manuscript. In response to your feedback, we have made every effort to address your concerns by providing further clarification on the experimental settings and adding more experiments under realistic scenarios.
>
> We would be very grateful for any additional feedback you may have on the revised manuscript and our responses. If there are any aspects that remain unclear, we are more than happy to provide further clarification.
>
> Thank you once again for your time and thoughtful review. We look forward to your response.
>
> Best regards,
>
> The Authors

---

### Official Review · Reviewer_yfrC · 2024-11-01

**Soundness:** 2
**Presentation:** 2
**Contribution:** 2
**Rating:** 5
**Confidence:** 3

**Summary:**

This paper proposes a novel attack-agnostic augmentation method to enhance the stealthiness and efficacy of existing poisoning attacks in federated learning (FL) systems. Unlike traditional poisoning methods that modify all model parameters uniformly, the proposed method targets a compact subnetwork, or "pill," which allows for a more subtle attack capable of evading detection. The approach involves three main stages: pill construction, pill poisoning, and pill injection. By focusing on critical model parameters, this method bypasses multiple state-of-the-art (SOTA) defenses in FL, as evidenced by experimental results showing significantly increased error rates under various conditions.

**Strengths:**

The proposed method introduces a unique attack strategy that optimizes stealth by selectively poisoning critical model parameters, making it less detectable by standard FL defenses.

**Weaknesses:**

The paper does not thoroughly address the scalability of the proposed attack in large-scale federated learning (FL) environments with diverse client devices and varying computational capabilities.

The dynamic pill search and injection stages could introduce significant computational overhead, especially in deep or complex models. This overhead could limit the attack's practicality in scenarios with frequent model updates or large FL models.

The paper’s experiments focus on a few specific model architectures, and it remains unclear whether the proposed pill-based poisoning technique can generalize across diverse FL model types, such as transformer-based models.

More baseline attacks should be considered.

**Questions:**

See the weakness.

---

> ### Comment · Reviewer_yfrC · 2024-11-27
>
> Thank you for the authors' replying. After reading the rebuttal, I decide to maintain my scores.

---

> ### Author Response · Authors · 2024-11-28
> **Response to Reviewer yfrC (Part 1)**
>
> Thanks for your insightful review and suggestions. Here are our detailed point-by-point feedbacks for your questions:
>
> > W1: The paper does not thoroughly address the scalability of the proposed attack in large-scale federated learning (FL) environments with diverse client devices and varying computational capabilities.
>
> To further demonstrate the effectiveness of our method in larger Federated Learning systems, we extend our experiments on the Fashion-MNSIT dataset with 100 clients. Here are the results using Trim Attack as the baseline attack:
> |     Attack     |   FedAvg  |  FLTrust  |   MKrum   |   Bulyan  |   Median  |    Trim   |    FLD    |
> |:--------------:|:---------:|:---------:|:---------:|:---------:|:---------:|:---------:|:---------:|
> |    No Attack   |   0.093   |   0.097   |   0.093   |   0.094   |   0.111   |   0.092   |   0.093   |
> |   Trim Attack  |   0.274   | **0.126** |   0.108   |   0.105   |   0.191   | **0.219** |   0.097   |
> | w/ Poison Pill | **0.336** |   0.101   | **0.901** | **0.281** | **0.272** |   0.122   | **0.518** |
>
> The results show a similar trend as observed in the 50-client system. Our method enables baseline attacks to successfully bypass four additional defenses, causing over 50% additional error rates in the global model. These findings further validate the effectiveness and generality of our approach in larger systems, when a single malicious client has fewer data samples.
>
> ---
>
> > W2: The dynamic pill search and injection stages could introduce significant computational overhead, especially in deep or complex models. This overhead could limit the attack's practicality in scenarios with frequent model updates or large FL models.
>
> Thank you for raising this issue. Due to the page limitation, we included the overhead analysis in Appendix D. This analysis demonstrates that the computational complexity of our pill search is much smaller than that of one round of local training. Therefore, the pill search stage introduces minimal additional overhead.
>
> Regarding the pill injection stage, we acknowledge that it may incur some additional computational costs. However, as shown in prior studies [1, 2], where post-adjustment of malicious updates is also employed, it is reasonable to assume that attackers could have access to more computational resources than a single benign client. This assumption aligns with the threat model used in our paper and most of the prior studies.
>
> ---
>
> > W3: The paper’s experiments focus on a few specific model architectures, and it remains unclear whether the proposed pill-based poisoning technique can be generalized across diverse FL model types, such as transformer-based models.
>
> To further demonstrate the effectiveness of our method on more model architectures, we test our method using VGG-11 net on the CIFAR-10 dataset. Here are the detailed results:
>
> Results with Trim Attack:
> |     Attack     |   FedAvg  |  FLTrust  |   MKrum   |   Bulyan  |   Median  |    Trim   |    DnC    |    FLD    |   Flame   |
> |:--------------:|:---------:|:---------:|:---------:|:---------:|:---------:|:---------:|:---------:|:---------:|:---------:|
> |    No Attack   |   0.319   |   0.328   |   0.338   |   0.324   |   0.330   |   0.337   |   0.315   |   0.334   |   0.336   |
> |   Trim Attack  |   0.431   |   0.323   |   0.422   |   0.428   | **0.434** | **0.432** |   0.340   | **0.339** |   0.362   |
> | w/ Poison Pill | **0.490** | **0.578** | **0.595** | **0.506** |   0.428   |   0.392   | **0.406** |   0.295   | **0.383** |
>
> Results with Sign-Flipping Attack:
> |        Attack        |   FedAvg  |  FLTrust  |   MKrum   |   Bulyan  |   Median  |    Trim   |    DnC    |    FLD    |   Flame   |
> |:--------------------:|:---------:|:---------:|:---------:|:---------:|:---------:|:---------:|:---------:|:---------:|:---------:|
> |       No Attack      |   0.319   |   0.328   |   0.338   |   0.324   |   0.330   |   0.337   |   0.315   |   0.334   |   0.336   |
> | Sign-flipping Attack | **0.897** |   0.335   |   0.336   |   0.329   |   0.353   |   0.386   |   0.341   |   0.316   |   0.367   |
> |    w/ Poison Pill    |   0.711   | **0.483** | **0.503** | **0.457** | **0.385** | **0.410** | **0.413** | **0.898** | **0.487** |
>
> The results demonstrate that our method consistently enhances the performance of both the Trim Attack and the Sign-Flipping Attack, outperforming their original versions in 14 out of 18 cases. These findings illustrate the effectiveness of our method with more complex model architectures, such as VGG-11. For transformer-based models, our method would also be applicable as long as a subnet exists that connects the input neurons to the output neurons.

---

> ### Author Response · Authors · 2024-11-28
> **Response to Reviewer yfrC (Part 2)**
>
> > W4: More baseline attacks should be considered.
>
> Thanks for the suggestion. We have included a new data-poisoning-based baseline attack, label-flipping attack, in comparison. It flips all the labels of the training data on malicious clients. Below are the results on the Fashion-MNIST dataset within a 50-client FL system:
> |     Attack     |   FedAvg  |  FLTrust  |   MKrum   |   Bulyan  |   Median  |    Trim   |    FLD    |
> |:--------------:|:---------:|:---------:|:---------:|:---------:|:---------:|:---------:|:---------:|
> |    No Attack   |   0.109   | **0.107** |   0.105   |   0.105   |   0.123   |   0.106   |   0.115   |
> |    LF Attack   | **0.960** | **0.107** |   0.095   |   0.105   |   0.116   |   0.115   |   0.096   |
> | w/ Poison Pill |   0.255   |   0.105   | **0.171** | **0.231** | **0.827** | **0.406** | **0.962** |
>
> The results demonstrate that our method enhances the Label-Flipping Attack to bypass five additional defenses compared to its original version. This illustrates the compatibility of our method with data-poisoning-based attacks.
>
> We also compared our method with Neurotoxin [3], an attack enhancement method designed for backdoor attacks. Below are the results on the IID Fashion-MNIST dataset within a 50-client FL system using Trim Attack:
> |     Attack     |   FedAvg  |  FLTrust  |   MKrum   |   Bulyan  |   Median  |    Trim   |    FLD    |
> |:--------------:|:---------:|:---------:|:---------:|:---------:|:---------:|:---------:|:---------:|
> |   Trim Attack  |   0.243   |   0.109   |   0.139   |   0.146   |   0.174   |   0.179   | **0.116** |
> | w/ Poison Pill | **0.618** | **0.576** | **0.638** | **0.284** | **0.453** | **0.219** |   0.115   |
> |  w/ Neurotoxin |   0.135   |   0.109   |   0.113   |   0.106   |   0.126   |   0.119   |   0.108   |
>
> We also evaluated the performance using the Sign-Flipping Attack with the same setting:
> |        Attack        |   FedAvg  |  FLTrust  |   MKrum   |   Bulyan  |   Median  |    Trim   |    FLD    |
> |:--------------------:|:---------:|:---------:|:---------:|:---------:|:---------:|:---------:|:---------:|
> | Sign-flipping Attack | **0.943** |   0.114   |   0.108   |   0.126   |   0.136   |   0.116   |   0.118   |
> |    w/ Poison Pill    |   0.667   |   0.115   | **0.764** | **0.379** | **0.523** | **0.314** | **0.646** |
> |     w/ Neurotoxin    |   0.710   | **0.147** |   0.105   |   0.103   |   0.106   |   0.105   |   0.110   |
>
> The results demonstrate that our method outperforms Neurotoxin in 12 out of 14 cases.
>
> These results further support the versatility and effectiveness of our method across a range of attack types.
>
> ---
>
> **References**
>
> 1. Shejwalkar, Virat, and Amir Houmansadr. "Manipulating the byzantine: Optimizing model poisoning attacks and defenses for federated learning." Network and Distributed System Security Symposium (NDSS). 2021.
> 2. Fang, Minghong, et al. "Local model poisoning attacks to {Byzantine-Robust} federated learning." 29th USENIX Security Symposium (USENIX Security 20). 2020.
> 3. Zhang, Zhengming, et al. "Neurotoxin: Durable backdoors in federated learning." International Conference on Machine Learning. 2022.

---

> ### Author Response · Authors · 2024-12-02
> **Kind Reminder from Authors**
>
> Dear Reviewer yfrC,
>
> We sincerely appreciate your valuable suggestions, which have significantly improved the quality of our manuscript. In response to your feedback, we have made every effort to address your concerns by adding further experiments on scalability and generalizability.
>
> We would be very grateful for any additional feedback you may have on the revised version and our responses. If there are any aspects that remain unclear, we are more than happy to provide further clarification.
>
> If our responses have adequately addressed your concerns, we kindly ask you to reconsider the score.
>
> Thank you once again for your time and thoughtful review. We look forward to your response and further discussion.
>
> Best regards,
>
> The Authors

---

### Official Review · Reviewer_98hg · 2024-11-02

**Soundness:** 3
**Presentation:** 3
**Contribution:** 2
**Rating:** 5
**Confidence:** 4

**Summary:**

This manuscript introduces an attack-agnostic augmentation technique designed to improve the stealth and effectiveness of existing poisoning attacks in Federated Learning (FL) systems. Experimental results demonstrate that, when enhanced by this method, FL poisoning attacks can evade state-of-the-art defenses, resulting in up to a 7-fold increase in error rates for current attack models.

**Strengths:**

• The manuscript is well-written and easy to follow.

• The study conducts extensive experiments to verify the effectiveness of the proposed method.

**Weaknesses:**

• The technical challenges and gaps in existing works are not adequately addressed. From the reviewer's perspective, this study merely adds an additional component that aims to identify 'critical neurons' in the network and applies existing attacks only to those neurons. The reviewer does not see the benefit of this extra component, as it may weaken the attack's effectiveness and increase computational overhead due to the identification process.

**Questions:**

• The 'Dynamic Pill Search' component is key point of the proposed method, but its details are missing from the main content. It is recommended to move these details from the appendix into the main text.

• Regarding the 'Pill Poisoning' design, it utilizes an ‘extra-trained model update’ as a reference model, aiming to make the generated malicious model update less divergent from the global model. This seems to contradict the original attack's goal, which is to maximize the difference between malicious and benign models. Therefore, the reviewer questions why this approach would outperform the original attacks in the FedAvg case.

• It is also suggested to evaluate the proposed method on targeted attacks.

**Details Of Ethics Concerns:**

Not applicable.

---

> ### Author Response · Authors · 2024-11-28
> **Response to Reviewer 98hg (Part 1)**
>
> Thanks for your insightful review. Here are our point-by-point responses:
>
> > W1: The technical challenges and gaps in existing works are not adequately addressed. From the reviewer's perspective, this study merely adds an additional component that aims to identify 'critical neurons' in the network and applies existing attacks only to those neurons. The reviewer does not see the benefit of this extra component, as it may weaken the attack's effectiveness and increase computational overhead due to the identification process.
>
> In our paper, we design a comprehensive pipeline that addresses key gaps in existing works and overcomes their limitations through the following contributions:
> 1. **Searching for a Stealthy but Effective Subnet (Pill)**:
> We propose a method to identify a stealthy yet effective subnet (i.e., the poison pill) within the global model. This search minimizes the scope of the attack to avoid broad and easily detectable modifications, ensuring both stealthiness and efficiency.
> 2. **Poisoning the Pill with Minimal Intrusion**:
> Our approach integrates seamlessly with existing baseline attacks by introducing minimal changes to their operation. This ensures high compatibility and minimal prerequisites for our method.
> 3. **Injecting the Poison Pill and Ensuring Integrity**:
> We carefully design the pill injection process to embed the malicious updates into the estimated benign updates. By gradually disconnecting the poison pill from other neurons in the global model, we also ensure the attacking integrity.
>
> Our method provides significant benefits compared to existing approaches:
> - **Enhanced Stealthiness:** The poison pill ensures that malicious updates bypass more robust defenses, making the attack less detectable.
> - **Improved Effectiveness:** By targeting only critical components, our method achieves higher attacking performance across diverse datasets, FL system settings, and model architectures.
> - **Broader Applicability:** Unlike existing studies, our method is effective even for untargeted attacks, further demonstrating its versatility.
>
> These contributions highlight the unique value of our approach, addressing gaps in the literature while maintaining a practical and scalable attack framework.
>
> ---
>
> > Q1: The 'Dynamic Pill Search' component is the key point of the proposed method, but its details are missing from the main content. It is recommended to move these details from the appendix into the main text.
>
> Thanks for your suggestion. We have moved more details about “Dynamic Pill Search” back to the main text in our revised manuscript.

---

> ### Author Response · Authors · 2024-11-28
> **Response to Reviewer 98hg (Part 2)**
>
> > Q2: Regarding the 'Pill Poisoning' design, it utilizes an ‘extra-trained model update’ as a reference model, aiming to make the generated malicious model update less divergent from the global model. This seems to contradict the original attack's goal, which is to maximize the difference between malicious and benign models. Therefore, the reviewer questions why this approach would outperform the original attacks in the FedAvg case.
>
> Thank you for the question. As shown in Figure 2 of our manuscript, attacks based on extra-trained model updates increase the similarity between the malicious update and the actual global update, enhancing the stealthiness of the attack. This works because the extra-trained model estimates the benign update in future rounds, which is not identical to the benign update in the current round. As a result, attacks crafted using such “future” model updates have less conflict with the current round’s benign update. Despite this, the malicious update remains effective because it is intentionally designed to oppose the direction of the model update trained from benign data.
>
> Regarding why our method outperforms baseline attacks in the FedAvg case, we attribute this to two key factors:
> 1. Stealthiness and Less Conflict with Benign Updates: Our method produces malicious updates that are stealthier and cause less conflict with benign updates. By leveraging the pill-based poisoning strategy and the extra-trained model update as the attack base, the effective poison pill is more likely to be entirely incorporated into the global model during the FedAvg aggregation. This contrasts with baseline attacks, where the malicious updates modify all parameters, often opposing the benign updates obviously. Such modifications by baseline attacks are more likely to be neutralized by the averaging operation in FedAvg, reducing their effectiveness.
> 2. Attacking Integrity via Gradual Disconnection: Our method gradually disconnects the poison pill from the model (Section 4.4), ensuring that the pill remains isolated from un-poisoned neurons in the final global model’s inference. This design preserves the integrity of the attack and ensures that the poisoned parameters remain effective. In contrast, baseline attacks often result in a partially poisoned global model, where the poisoned neurons are influenced by un-poisoned neurons during inference, reducing the overall attack effectiveness.
>
> ---
>
> > Q3: It is also suggested to evaluate the proposed method on targeted attacks.
>
> Thank you for the suggestion. Most targeted attacks in federated learning (FL) are based on data poisoning. We acknowledge that our current evaluation lacks baseline attacks based on such data poisoning methods. To address this and provide further evidence of the effectiveness of our method, we have included a new data-poisoning-based targeted attack in our comparison: Label-Flipping Attack.
>
> Label-flipping is a simple yet effective targeted attack in FL, and it is also one of the least stealthy data-poisoning-based attacks. To make the evaluation more challenging, we configured the attacker to flip all the labels of the training data on malicious clients, making the label-flipping attack even less stealthy. Below are the error rate results on the Fashion-MNIST dataset within a 50-client FL system. Since we flipped all the labels on malicious clients, a high error rate would be expected.
> |     Attack     |   FedAvg  |  FLTrust  |   MKrum   |   Bulyan  |   Median  |    Trim   |    FLD    |
> |:--------------:|:---------:|:---------:|:---------:|:---------:|:---------:|:---------:|:---------:|
> |    No Attack   |   0.109   | **0.107** |   0.105   |   0.105   |   0.123   |   0.106   |   0.115   |
> |    LF Attack   | **0.960** | **0.107** |   0.095   |   0.105   |   0.116   |   0.115   |   0.096   |
> | w/ Poison Pill |   0.255   |   0.105   | **0.171** | **0.231** | **0.827** | **0.406** | **0.962** |
>
> The results demonstrate that our method enhances the Label-Flipping Attack to bypass five additional defenses compared to its original version. This illustrates the compatibility of our method with data-poisoning-based and targeted attacks.

---

> ### Author Response · Authors · 2024-12-02
> **Kind Reminder from Authors**
>
> Dear Reviewer 98hg,
>
> We sincerely appreciate your valuable suggestions, which have significantly improved the quality of our manuscript. In response to your feedback, we have made every effort to address your concerns by adding more details about "Dynamic Pill Search" into our revised main text and presenting more detailed results with targeted attacks.
>
> We would be grateful for any further feedback you may have on the revised version and our responses. If there are any aspects that remain unclear, we are more than willing to provide additional clarification.
>
> If our responses have adequately addressed your concerns, we kindly ask you to reconsider the score.
>
> Thank you once again for your time and thoughtful review. We are looking forward to your response and further discussion.
>
> Best regards,
>
> The Authors

---

### Official Review · Reviewer_L8wH · 2024-11-04

**Soundness:** 2
**Presentation:** 2
**Contribution:** 2
**Rating:** 3
**Confidence:** 5

**Summary:**

This paper introduces an attack-agnostic augmentation method to bypass existing FL defenses and increase error rates. The high-level idea of this method is to constrain the poisoned updates into a sub-network instead of poisoning the whole model architecture. Via empirical experiments, this method is shown to bypass existing FL defenses and can be used as an add-on to four model poisoning attacks.

**Strengths:**

- This method can be combined with various poisoning attacks in FL and the idea of using sub-network sounds interesting
- The experiments are conducted on nine defenses and three datasets and show some improvement in increasing error rates
- This paper considers the stealthiness and effectiveness of the attack at the same time

**Weaknesses:**

1. The writing of this paper needs extensive revision for the following reasons: (i) the method is written using mostly oral sentences, lacking formal equations and (ii) the intuition and contribution of each sub-component are unclear. For example, in “**Designing Pill Blueprint”,** how the most important parameters are determined and what is its contribution to the overall method is not clearly stated.
2. This paper lacks an ablation study to investigate the effectiveness of each component including (i) the percentage of data that the adversary holds per total client data, (ii) the effect of SimAdjust and DisAdjust in L16-17, Algo. 1.
3. Results in Table 2 do not sound convincing. Except for Sign-flipping Attack, the error rates of baseline attacks are very low, i.e., 10-20% even with FedAvg (no-defense), and FLD defense even increases the error rates. On the other hand, in the original paper from Fang et al. 2020, the error rates can reach up to 80% in some cases. Can the authors explain these?
4. Related attacks focusing on model poisoning to increase stealthiness and effectiveness such as PDG, Model Replacement, Constrain-and-Scale, and Neurotoxin are not compared or discussed.

[1] Bagdasaryan, Eugene, et al. "How to backdoor federated learning." *International conference on artificial intelligence and statistics*. PMLR, 2020.

[2]  Bhagoji, Arjun Nitin, et al. "Analyzing federated learning through an adversarial lens." *International conference on machine learning*. PMLR, 2019.

[3] Zhang, Zhengming, et al. "Neurotoxin: Durable backdoors in federated learning." *International Conference on Machine Learning*. PMLR, 2022.

**Questions:**

1. This method seems over-complicated with extensive training steps. How is the computational time of this attack compared to others from benign clients? whether it can introduce too long a waiting tie and raise suspicious from the server.
2. How to ensure the fidelity of the estimated benign update, since it is heavily depending on the quality and distribution of the data from the adversarial clients? What is the assumption about the data that the adversary has across compromised clients?
3. In Line 257,  what is the contribution of the starting point defined by an attacker on the overall performance since the error can be propagated to the next layers if the first neuron on the first layer is not the optimal one? This procedure and why does it work is unclear to me.
What is the indication of these fill’s neurons, and their role on the network?
This discussion should be presented in the main manuscript.
4. The number of clients is smaller than the existing FL attack setting, which may lead to a large amount of data that an attacker has, could authors show this method can work with larger systems and smaller amounts of data from attackers?
5. “Malicious 20%” is too strong and unrealistic, whereas practical attacks often use 1-5% of the malicious clients. Whether or not this attack can work with smaller attacker numbers?
6. Can this method be used with backdoor attacks?
7. Can this method work with more complex model architecture such as ResNet/VGG?

---

> ### Author Response · Authors · 2024-11-28
> **Response to Reviewer L8wH (Part 1)**
>
> Thanks for your insightful review. Here are our detailed point-by-point feedbacks for your questions:
>
> > W1: The writing of this paper needs extensive revision for the following reasons: (i) the method is written using mostly oral sentences, lacking formal equations and (ii) the intuition and contribution of each sub-component are unclear. For example, in “Designing Pill Blueprint”, how the most important parameters are determined and what is its contribution to the overall method is not clearly stated.
>
> Thanks for your suggestions on the writing. Due to the page limits in the initial submission, we presented a detailed description of the pill search algorithm in Appendix D. However, we understand the importance of presenting such critical details in the main text. In the revised version, we have moved the key aspects of the algorithm back to the main text to enhance accessibility and clarity for readers.
>
> ---
>
> > W2: This paper lacks an ablation study to investigate the effectiveness of each component including (i) the percentage of data that the adversary holds per total client data, (ii) the effect of SimAdjust and DisAdjust in L16-17, Algo. 1.
>
> (i) In our paper, we have presented the results with 20% malicious clients (Table 2 in the main text) and 10% malicious clients (Table 3 in the main text), where the adversary controls 20% and 10% of the total client data, respectively. These percentages represent commonly used settings in prior research. Additionally, we have tested these two configurations across multiple datasets (MNIST, Fashion-MNIST, and CIFAR-10), data distributions (IID and non-IID), models (4-layer CNN, AlexNet, and VGG-11), and total client numbers (30, 50, and 100).
>
> (ii) During the rebuttal phase, we conducted a detailed ablation study to evaluate the effects of SimAdjust and DistAdjust in our method. Below, we provide the error rates of the Trim Attack with different settings of our method on the IID Fashion-MNIST dataset within a 50-client FL system:
> |                   Attack                   |   FedAvg  |  FLTrust  |   MKrum   |   Bulyan  |   Median  |    Trim   |    FLD    |
> |:------------------------------------------:|:---------:|:---------:|:---------:|:---------:|:---------:|:---------:|:---------:|
> |                 Trim Attack                |   0.243   |   0.109   |   0.139   |   0.146   |   0.174   |   0.179   |   0.116   |
> |  w/ Poison Pill w/ SimAdjust w/ DistAdjust | **0.618** | **0.576** | **0.638** | **0.284** | **0.453** |   0.219   |   0.115   |
> | w/ Poison Pill w/o SimAdjust w/ DistAdjust |   0.317   |   0.105   |   0.364   |   0.247   |   0.368   |   0.136   | **0.208** |
> | w/ Poison Pill w/ SimAdjust w/o DistAdjust |   0.554   |   0.104   |   0.122   |   0.108   |   0.429   | **0.284** |   0.119   |
>
> The results demonstrate that simultaneously using both SimAdjust and DistAdjust outperforms using only SimAdjust or DistAdjust in 5 out of 7 cases. While using just one adjustment method may surpass the combined approach in one or two specific scenarios, it does not ensure consistent bypassing effectiveness across diverse defenses. This highlights the necessity of the combined adjustment in our method, which leverages the complementary strengths of both SimAdjust and DistAdjust to achieve superior performance.

---

> ### Author Response · Authors · 2024-11-28
> **Response to Reviewer L8wH (Part 2)**
>
> > W3: Results in Table 2 do not sound convincing. Except for Sign-flipping Attack, the error rates of baseline attacks are very low, i.e., 10-20% even with FedAvg (no-defense), and FLD defense even increases the error rates. On the other hand, in the original paper from Fang et al. 2020, the error rates can reach up to 80% in some cases. Can the authors explain these?
>
> Thank you for raising this issue. The low error rates of baseline attacks can be attributed to two factors:
>
> 1. Practical Threat Model: In our work, we adopt a practical threat model with a relatively low proportion of malicious clients and a partial knowledge setting. While this approach is more realistic, it also poses greater challenges for attackers. Notably, in the original paper by Fang et al. (2020), the attack performance is significantly lower in the partial knowledge setting as well.
>
> 2. Exclusion of Weak Baselines: Instead of Krum, we utilize Multi-Krum, which excludes a relatively weak baseline defense commonly used in prior studies [1,2,3]. As a result, the baseline defenses in our paper are more robust compared to prior studies [1,2,3], causing existing baseline attacks to appear less effective across our evaluations. It is worth noting that in follow-up methods like FLTrust, the attacks proposed by Fang et al. (2020) exhibit similar performance to our implementation, except in cases using Krum (which we replaced with Multi-Krum in our study).
>
> Regarding the undesirable performance of FLDetector, the reason could be not all baseline attacks were tested in its original paper (e.g., Min-Max Attack). Consequently, FLDetector may fail to defend against such untested attacks. However, in our results, FLDetector demonstrates robustness against attacks evaluated in its original paper, such as the Sign-Flipping Attack and Trim Attack.
>
> ---
>
> > W4: Related attacks focusing on model poisoning to increase stealthiness and effectiveness such as PDG, Model Replacement, Constrain-and-Scale, and Neurotoxin are not compared or discussed.
>
> Thank you for your suggestion. While these methods are primarily designed for backdoor attacks, which may not be fully applicable to the untargeted attacks used in our experiments, we agree that certain aspects of these methods could be adapted to the setting in our paper. To provide a more comprehensive evaluation, we compared our method with Neurotoxin, as suggested by the reviewer. Below are the results on the IID Fashion-MNIST dataset within a 50-client FL system using Trim Attack:
> |     Attack     |   FedAvg  |  FLTrust  |   MKrum   |   Bulyan  |   Median  |    Trim   |    FLD    |
> |:--------------:|:---------:|:---------:|:---------:|:---------:|:---------:|:---------:|:---------:|
> |   Trim Attack  |   0.243   |   0.109   |   0.139   |   0.146   |   0.174   |   0.179   | **0.116** |
> | w/ Poison Pill | **0.618** | **0.576** | **0.638** | **0.284** | **0.453** | **0.219** |   0.115   |
> |  w/ Neurotoxin |   0.135   |   0.109   |   0.113   |   0.106   |   0.126   |   0.119   |   0.108   |
>
> We also evaluated the performance using the Sign-Flipping Attack with the same setting:
> |        Attack        |   FedAvg  |  FLTrust  |   MKrum   |   Bulyan  |   Median  |    Trim   |    FLD    |
> |:--------------------:|:---------:|:---------:|:---------:|:---------:|:---------:|:---------:|:---------:|
> | Sign-flipping Attack | **0.943** |   0.114   |   0.108   |   0.126   |   0.136   |   0.116   |   0.118   |
> |    w/ Poison Pill    |   0.667   |   0.115   | **0.764** | **0.379** | **0.523** | **0.314** | **0.646** |
> |     w/ Neurotoxin    |   0.710   | **0.147** |   0.105   |   0.103   |   0.106   |   0.105   |   0.110   |
>
> The results demonstrate that our method outperforms Neurotoxin in 12 out of 14 cases. This highlights that directly transferring existing methods designed for backdoor attacks may not yield consistent effectiveness when applied to untargeted attack scenarios. The results further validate the robustness of our approach.

---

> ### Author Response · Authors · 2024-11-28
> **Response to Reviewer L8wH (Part 3)**
>
> > Q1: This method seems over-complicated with extensive training steps. How is the computational time of this attack compared to others from benign clients? whether it can introduce too long a waiting tie and raise suspicious from the server.
>
> In prior studies [1, 2, 3, 5], attackers are generally assumed to possess greater capabilities, including access to more data and computational resources. Consistent with these settings, our method assumes that attackers have computational resources exceeding those of benign clients.
>
> Additionally, the extra training required by our method can be executed in parallel with the local training on each malicious client. This parallelization ensures that the attack does not introduce significant delays and minimizes the risk of raising suspicion from the server.
>
> Regarding the computational overhead of the pill search, we have provided a detailed complexity analysis in Appendix D, which outlines the efficiency of our method and its practicality within the assumed threat model.
>
> ---
>
> > Q2: How to ensure the fidelity of the estimated benign update, since it is heavily depending on the quality and distribution of the data from the adversarial clients? What is the assumption about the data that the adversary has across compromised clients?
>
> The intuition behind utilizing the estimated benign update is that the aggregated update is more similar to the global update than any individual update trained solely on data from a specific malicious client. This property ensures that the estimated benign update better reflects the overall data distribution. Additionally, we do not rely on strong or restrictive assumptions about the estimated benign update, which makes our approach broadly applicable across different settings.
>
> Regarding the assumptions about data distribution, we assign data to both benign and malicious clients using the same IID or non-IID distribution rules, as implemented in prior studies [1,2,3] and detailed in Appendix G. This ensures a fair and controlled comparison, avoiding any undue advantage to attackers while maintaining consistency with established benchmarks.
>
> ---
>
> > Q3: In Line 257, what is the contribution of the starting point defined by an attacker on the overall performance since the error can be propagated to the next layers if the first neuron on the first layer is not the optimal one? This procedure and why does it work is unclear to me. What is the indication of these fill’s neurons, and their role on the network? This discussion should be presented in the main manuscript.
>
> The goal of the pill search is to identify a pill that provides a shortcut from the selected starting point to all output neurons. While an individual neuron in the pill does not hold significant meaning on its own, the entire set of neurons in the pill collectively represents a stealthy and effective target neuron set for the attacker.
>
> The selection of the starting point has minimal influence on the effectiveness of our method. Regardless of the chosen starting point, our approach reliably identifies an effective pill using the "approximate max pill search" algorithm, as detailed in Appendix D. When no specific requirement is imposed, the starting point is selected randomly. This design choice reduces the trackability of the poison pill, as selecting a starting point based on a fixed rule could allow defenses to easily identify the starting point and subsequently discover the pill.
>
> Due to the page constraints of the initial submission, we did not include these details in the main text. However, we recognize the importance of this discussion and will incorporate additional details in the revised manuscript’s main text to clarify the procedure and the roles of the pill’s neurons in the network.

---

> ### Author Response · Authors · 2024-11-28
> **Response to Reviewer L8wH (Part 4)**
>
> > Q4: The number of clients is smaller than the existing FL attack setting, which may lead to a large amount of data that an attacker has, could authors show this method can work with larger systems and smaller amounts of data from attackers?
>
> Thank you for your suggestion. We have further evaluated our method using the Trim Attack in a 100-client system on the Fashion-MNIST dataset. The results are as follows:
> |     Attack     |   FedAvg  |  FLTrust  |   MKrum   |   Bulyan  |   Median  |    Trim   |    FLD    |
> |:--------------:|:---------:|:---------:|:---------:|:---------:|:---------:|:---------:|:---------:|
> |    No Attack   |   0.093   |   0.097   |   0.093   |   0.094   |   0.111   |   0.092   |   0.093   |
> |   Trim Attack  |   0.274   | **0.126** |   0.108   |   0.105   |   0.191   | **0.219** |   0.097   |
> | w/ Poison Pill | **0.336** |   0.101   | **0.901** | **0.281** | **0.272** |   0.122   | **0.518** |
>
> The results show a similar trend as observed in the 50-client system. Our method enables baseline attacks to successfully bypass four additional defenses, causing over 50% additional error rates in the global model. These findings further validate the effectiveness and generality of our approach in larger systems.
>
> ---
>
> > Q5: “Malicious 20%” is too strong and unrealistic, whereas practical attacks often use 1-5% of the malicious clients. Whether or not this attack can work with smaller attacker numbers?
>
> We have presented results with 10% malicious clients in Table 3 of the main text and Table 10 in Appendix L. These results show that our method achieves consistent attack augmentation performance under both 10% and 20% malicious client proportions.
>
> While studies on backdoor attacks often consider lower malicious client proportions (e.g., 1–10%), papers focusing on untargeted attacks [1, 2, 3] typically use higher proportions, ranging from 10% to 40%, to ensure attack effectiveness. In line with these prior studies, we adopt two relatively low malicious client proportions, 10% and 20%, for our experiments.
>
> ---
>
> > Q6: Can this method be used with backdoor attacks?
>
> In this paper, we primarily focus on untargeted model poisoning attacks, which involve directly modifying updates on malicious clients. These attacks are less stealthy and harder to camouflage compared to backdoor attacks, making them suitable for demonstrating the more evident effectiveness of our method. Since our method shows high performance in camouflaging untargeted attacks, it can be reasonably inferred that it would also perform effectively with more stealthy backdoor attacks. Furthermore, as part of the core theory of our method is derived from the SRA attack [4], which is specifically designed for backdoor attacks, our method is inherently compatible with existing backdoor attack strategies.
>
> While we acknowledge the absence of data-poisoning-based baseline attacks in our initial experiments, we have included a new label-flipping attack in the comparison, which flips all the labels of the training data on malicious clients. Below are the results on the Fashion-MNIST dataset within a 50-client FL system:
> |     Attack     |   FedAvg  |  FLTrust  |   MKrum   |   Bulyan  |   Median  |    Trim   |    FLD    |
> |:--------------:|:---------:|:---------:|:---------:|:---------:|:---------:|:---------:|:---------:|
> |    No Attack   |   0.109   | **0.107** |   0.105   |   0.105   |   0.123   |   0.106   |   0.115   |
> |    LF Attack   | **0.960** | **0.107** |   0.095   |   0.105   |   0.116   |   0.115   |   0.096   |
> | w/ Poison Pill |   0.255   |   0.105   | **0.171** | **0.231** | **0.827** | **0.406** | **0.962** |
>
> The results demonstrate that our method enhances the Label-Flipping Attack to bypass five additional defenses compared to its original version. This illustrates the compatibility of our method with data-poisoning-based attacks, including backdoor attacks, further supporting its versatility and effectiveness across a range of attack types.

---

> ### Author Response · Authors · 2024-11-28
> **Response to Reviewer L8wH (Part 5)**
>
> > Q7: Can this method work with more complex model architecture such as ResNet/VGG?
>
> Thank you for the question. We evaluated our method using the VGG-11 model on the CIFAR-10 dataset with both the Trim Attack and the less stealthy Sign-Flipping Attack. The results are as follows:
>
> Results with Trim Attack:
> |     Attack     |   FedAvg  |  FLTrust  |   MKrum   |   Bulyan  |   Median  |    Trim   |    DnC    |    FLD    |   Flame   |
> |:--------------:|:---------:|:---------:|:---------:|:---------:|:---------:|:---------:|:---------:|:---------:|:---------:|
> |    No Attack   |   0.319   |   0.328   |   0.338   |   0.324   |   0.330   |   0.337   |   0.315   |   0.334   |   0.336   |
> |   Trim Attack  |   0.431   |   0.323   |   0.422   |   0.428   | **0.434** | **0.432** |   0.340   | **0.339** |   0.362   |
> | w/ Poison Pill | **0.490** | **0.578** | **0.595** | **0.506** |   0.428   |   0.392   | **0.406** |   0.295   | **0.383** |
>
> Results with Sign-Flipping Attack:
> |        Attack        |   FedAvg  |  FLTrust  |   MKrum   |   Bulyan  |   Median  |    Trim   |    DnC    |    FLD    |   Flame   |
> |:--------------------:|:---------:|:---------:|:---------:|:---------:|:---------:|:---------:|:---------:|:---------:|:---------:|
> |       No Attack      |   0.319   |   0.328   |   0.338   |   0.324   |   0.330   |   0.337   |   0.315   |   0.334   |   0.336   |
> | Sign-flipping Attack | **0.897** |   0.335   |   0.336   |   0.329   |   0.353   |   0.386   |   0.341   |   0.316   |   0.367   |
> |    w/ Poison Pill    |   0.711   | **0.483** | **0.503** | **0.457** | **0.385** | **0.410** | **0.413** | **0.898** | **0.487** |
>
> The results demonstrate that our method consistently enhances the performance of both the Trim Attack and the Sign-Flipping Attack, outperforming their original versions in 14 out of 18 cases. These findings illustrate the effectiveness of our method with more complex model architectures, such as VGG-11.
>
> ---
>
> **References**
>
> 1. Shejwalkar, Virat, and Amir Houmansadr. "Manipulating the byzantine: Optimizing model poisoning attacks and defenses for federated learning." Network and Distributed System Security Symposium (NDSS). 2021.
> 2. Fang, Minghong, et al. "Local model poisoning attacks to {Byzantine-Robust} federated learning." 29th USENIX Security Symposium (USENIX Security 20). 2020.
> 3. Cao, Xiaoyu, et al. "FLTrust: Byzantine-robust Federated Learning via Trust Bootstrapping." Network and Distributed System Security Symposium (NDSS). 2021.
> 4. Qi, Xiangyu, et al. "Towards practical deployment-stage backdoor attack on deep neural networks." IEEE/CVF Conference on Computer Vision and Pattern Recognition (CVPR). 2022.
> 5. Zhang, Kaiyuan, et al. "FLIP: A Provable Defense Framework for Backdoor Mitigation in Federated Learning." International Conference on Learning Representations (ICLR). 2023

---

> ### Author Response · Authors · 2024-12-02
> **Kind Reminder from Authors**
>
> Dear Reviewer L8wH,
>
> We sincerely appreciate your valuable suggestions, which have significantly improved the quality of our manuscript. In response to your feedback, we have made every effort to address your concerns by providing a more detailed ablation study and additional experimental results on generalizability, included in the newly added Appendix P to U.
>
> We would be grateful for any further feedback you may have on the revised version and our responses. If there are any aspects that remain unclear, we are more than willing to provide additional clarification.
>
> If our responses have adequately addressed your concerns, we kindly ask that you reconsider the score.
>
> Thank you once again for your time and thoughtful review. We are looking forward to your response and further discussion.
>
> Best regards,
>
> The Authors

---

### Author Response · Authors · 2024-11-28
**General Response**

We thank all the reviewers for your thoughtful comments! We are glad that the reviewers found our paper proposes a novel idea with a comprehensive evaluation. We also thank you for your appreciation of our method’s robustness and paper presentation.

To further address your concerns, we conducted more experiments and provided more detailed evidence to support our proposed method. Here is a summary of the supplementary information provided in the rebuttal materials:

1. We test our method using a more complex VGG-11 model.
2. We test our method in a 100-client system, where each malicious client has fewer data samples.
3. We test our method with the label-flipping attack, which is a data-poisoning-based baseline attack.
4. We add a comparison between our method and the Neurotoxin [1] suggested by the reviewer.
5. We provide a detailed ablation study of the effectiveness of each component in the pill adjustment stage.
6. We test our method with a setting where the malicious clients are gradually decreasing in order to evaluate our method in a more real-world scenario.
7. We further polish our manuscript to improve the clarity and logical flow of our paper. The revised contents are highlighted in blue.

We hope our responses address your concerns and we are looking forward to your further feedback. Thank you again for your comments and recognition of our work.

[1] Zhang, Zhengming, et al. "Neurotoxin: Durable backdoors in federated learning." International Conference on Machine Learning. 2022.

---

### Author Response · Authors · 2024-12-02
**Kind Reminder from Authors**

Dear Reviewers,

We sincerely appreciate the time and effort you have invested in reviewing our manuscript. As the discussion period is nearing its close, we wanted to kindly remind you that we remain available to provide further clarification on any points that may still be unclear regarding our manuscript or rebuttal materials.

We are deeply grateful for your thoughtful feedback and would be happy to assist with any additional questions.

Thank you once again for your valuable contributions.

Best regards,

The Authors

---

### Meta-Review · Area_Chair_yX6q · 2024-12-11

**Metareview:**

This paper proposes a data augmentation technique called Poison Pill to help various backdoor attacks on federated learning (FL) evade existing defenses. By demonstrating its effectiveness across different scenarios, the work highlights the vulnerability of current FL systems to advanced poisoning strategies like the proposed one. Although one reviewer was very positive about the paper, other reviewers raised major concerns regarding the presentation, clarity of the proposed methodology, experimental setting, and the scale of the experiments. While the authors have addressed some of these concerns with new results, several issues remain unresolved. In its current form, the paper does not meet ICLR's expectations. I recommend the authors continue refining the paper for future submissions.

**Additional Comments On Reviewer Discussion:**

None

---

### Decision · Program_Chairs · 2025-01-22

Reject